# Predictions and experimental tests of a new biophysical model of the mammalian respiratory oscillator

Ryan S Phillips[1,2†], Hidehiko Koizumi[3†], Yaroslav I Molkov[4,5], Jonathan E Rubin[1,2], Jeffrey C Smith[3]*

[1]Department of Mathematics, University of Pittsburgh, Pittsburgh, United States; [2]Center for the Neural Basis of Cognition, Pittsburgh, United States; [3]Cellular and Systems Neurobiology Section, NINDS, NIH, Bethesda, United States; [4]Department of Mathematics and Statistics, Georgia State University, Atlanta, United States; [5]Neuroscience Institute, Georgia State University, Atlanta, United States

**Abstract** Previously our computational modeling studies (Phillips et al., 2019) proposed that neuronal persistent sodium current ($I_{NaP}$) and calcium-activated non-selective cation current ($I_{CAN}$) are key biophysical factors that, respectively, generate inspiratory rhythm and burst pattern in the mammalian preBötzinger complex (preBötC) respiratory oscillator isolated in vitro. Here, we experimentally tested and confirmed three predictions of the model from new simulations concerning the roles of $I_{NaP}$ and $I_{CAN}$: (1) $I_{NaP}$ and $I_{CAN}$ blockade have opposite effects on the relationship between network excitability and preBötC rhythmic activity; (2) $I_{NaP}$ is essential for preBötC rhythmogenesis; and (3) $I_{CAN}$ is essential for generating the amplitude of rhythmic output but not rhythm generation. These predictions were confirmed via optogenetic manipulations of preBötC network excitability during graded $I_{NaP}$ or $I_{CAN}$ blockade by pharmacological manipulations in slices in vitro containing the rhythmically active preBötC from the medulla oblongata of neonatal mice. Our results support and advance the hypothesis that $I_{NaP}$ and $I_{CAN}$ mechanistically underlie rhythm and inspiratory burst pattern generation, respectively, in the isolated preBötC.

*For correspondence:
smithj2@ninds.nih.gov

†These authors contributed equally to this work

## Editor's evaluation

This paper tests hypotheses for the role of INaP and ICAN in the preBötC, the region of the brainstem that generates inspiratory breathing rhythm, using optogenetic manipulation of local preBötC excitability and pharmacologic blockade of INaP and ICAN and tests resulting predictions about these currents using computational simulation. The paper will be of interest to respiratory researchers and all those interested in neuronal rhythm generation.

## Introduction

The cellular and circuit biophysical mechanisms generating the breathing rhythm critical for life in mammals have been under investigation for decades without clear resolution (*Richter and Smith, 2014*; *Del Negro et al., 2018*; *Ramirez and Baertsch, 2018*). The roles of various transmembrane ionic currents in this process have received considerable attention, including our recent updated computational modeling (*Phillips et al., 2019*) of the biophysical mechanisms operating in the mammalian respiratory oscillator within the brainstem preBötzinger complex (preBötC), which contains essential rhythmogenic neuronal circuits.

Simulations from our updated model of preBötC excitatory circuits support distinct functional roles for cellular-level Na$^+$- and Ca$^{2+}$-based biophysical mechanisms that have been of intense experimental and theoretical interest for understanding the operation of preBötC circuits (*Phillips et al., 2019*). These mechanisms rely on a slowly inactivating neuronal persistent sodium current (I$_{NaP}$) and a calcium-activated non-selective cation current (I$_{CAN}$) mediated by transient receptor potential M4 (TRPM4) channels coupled to intracellular calcium dynamics. The model proposes how these cellular-level mechanisms operating in excitatory circuits can underlie two critical functions of preBötC circuits: generation of the inspiratory rhythm and regulation of the amplitude of inspiratory population activity. In essence, the model advances the concepts that (1) a subset of excitatory circuit neurons, whose rhythmic bursting in the isolated preBötC in vitro is critically dependent on I$_{NaP}$, forms an excitatory neuronal kernel for rhythm generation, and (2) excitatory synaptic drive from the rhythmogenic kernel population is critically amplified by I$_{CAN}$ activation in a recruited subset of preBötC excitatory neurons to generate overall population activity of sufficient amplitude to induce downstream excitatory circuits to produce inspiratory motor output.

While these concepts have some experimental support (see discussions and model-experimental data comparisons in *Phillips et al., 2019*), the mechanisms incorporated in the model and their predicted effects on circuit behavior require more extensive experimental testing. Indeed, the basic rhythmogenic I$_{NaP}$-based biophysical mechanism represented by the model remains controversial (*Del Negro et al., 2018*). The concept that activation of I$_{CAN}$ is largely due to synaptically activated sources of neuronal Ca$^{2+}$ flux, such that I$_{CAN}$ tunes inspiratory burst amplitude in proportion to the level of excitatory synaptic current from the kernel, also requires additional experimental support. In the present modeling and experimental study, to build on our previous work (*Phillips et al., 2019*), we have performed experimental tests on the rhythmically active in vitro network in slices from neonatal transgenic mice using a combination of electrophysiological analyses, pharmacological perturbations of I$_{NaP}$ or I$_{CAN}$, and optogenetic manipulations of the preBötC excitatory (glutamatergic) population involved. Our new comparisons of model simulations and experimental data provide further evidence for the predictive power and validity of the model, while also indicating some limitations.

## Results
### Model concepts and predictions
The following features of the model outputs from new simulations (*Figure 1*) formed the basis for the design of experiments for model simulation-experimental data comparisons in the current study. Simulations were performed with the model specified in *Phillips et al., 2019*, with modifications as described in Materials and methods.

1. The inspiratory rhythm in vitro is generated by a functionally distinct subpopulation of preBötC excitatory neurons with subthreshold-activating I$_{NaP}$ that confers voltage-dependent rhythmic burst frequency control, providing a mechanism for frequency tuning by the regulation of baseline membrane potentials. Progressively depolarizing/hyperpolarizing neurons across the population by varying an applied current (I$_{App}$) increases/decreases population burst frequency over a wide dynamic range defined by the frequency tuning curve for the population (i.e. the relationship between applied current/baseline membrane potential and network bursting frequency).
2. Reducing neuronal persistent sodium conductance (g$_{NaP}$) decreases the population-level bursting frequency and alters the voltage-dependent rhythmogenic behavior with a shift in the frequency tuning curve such that the frequency dynamic range is reduced, consistent with previous simulations (*Phillips and Rubin, 2019*). Population amplitude is not as strongly reduced by decreases in g$_{NaP}$. Rhythm generation can be terminated at a sufficiently low level of g$_{NaP}$ under baseline conditions of network excitability in vitro.
3. Reducing neuronal calcium-activated non-selective cation conductance [g$_{CAN}$] very strongly decreases population activity amplitude and, in contrast to reducing g$_{NaP}$, has little effect on population bursting frequency under baseline conditions, but strongly augments bursting frequency due to a shift in the frequency tuning curve that extends the frequency range at higher levels of population depolarization.

Model simulations showing the relations between I$_{App}$, population burst frequency, and burst amplitude at different levels of g$_{NaP}$ and g$_{CAN}$ are displayed in *Figure 1*, which illustrate these basic predictions of the model. Importantly, reducing the synaptic connection probability does not qualitatively

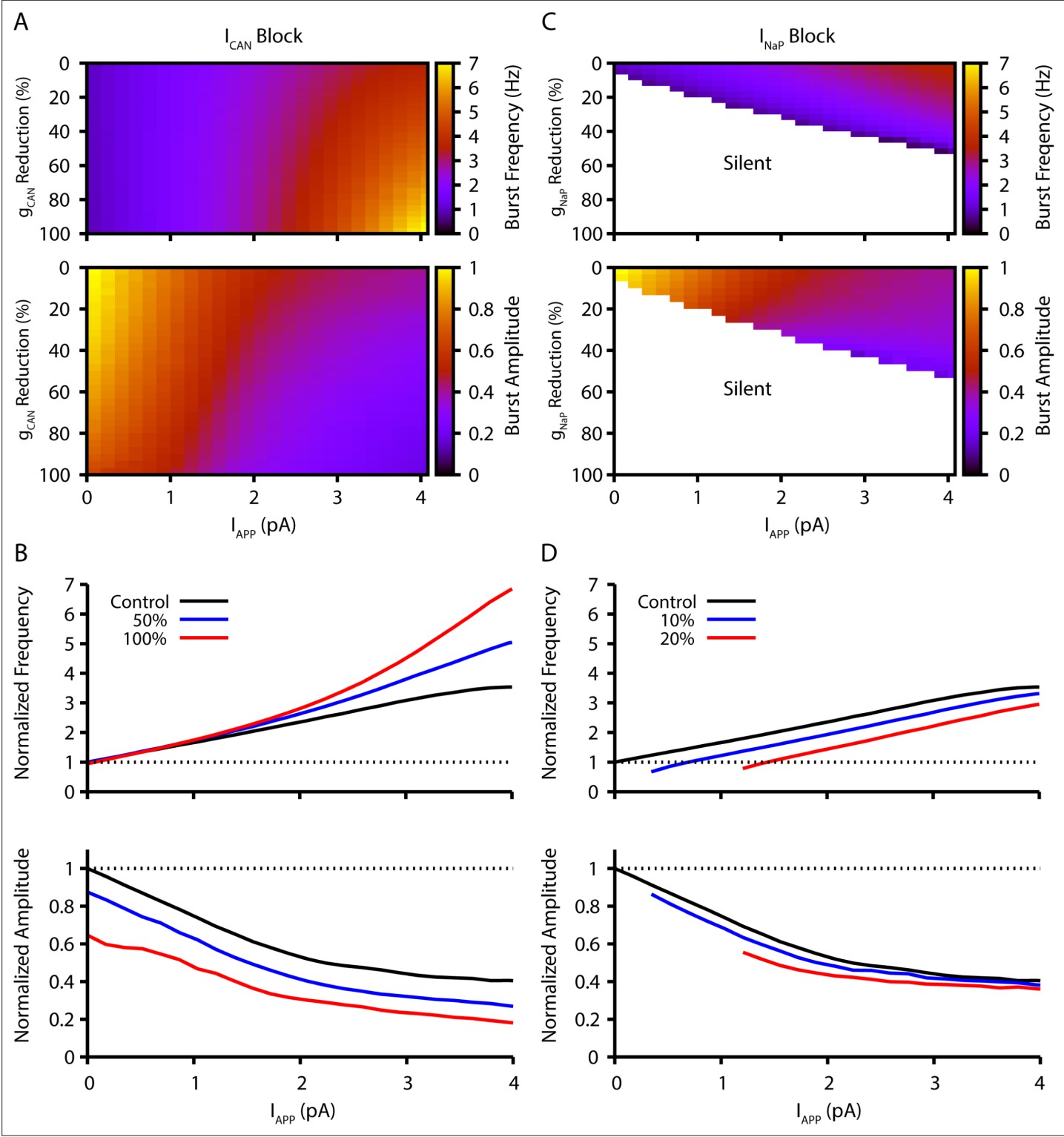

**Figure 1.** Model simulation predictions for the relationships between applied current ($I_{App}$) and population burst frequency and amplitude of synaptically coupled excitatory neurons (N=100) incorporating neuronal persistent sodium current ($I_{NaP}$) and calcium-activated non-selective cation current ($I_{CAN}$) over a range of conductances (calcium-activated non-selective cation conductance [$g_{CAN}$], neuronal persistent sodium conductance [$g_{NaP}$]). Pharmacological block of $I_{NaP}$ and $I_{CAN}$ is simulated by a percent reduction of their respective conductances. (**A & B**) Model parameter space plots color-coded from simulations show effects on frequency (upper panel in **A**) and amplitude (lower panel in **A**) of reductions in $g_{CAN}$, and in (**B**) at fixed levels of reductions (50%, 100%) in $g_{CAN}$ from initial control values over a wide range of $I_{App}$. (**C & D**) Parameter space plots showing effects in (**C**) of reductions in $g_{NaP}$ on

*Figure 1 continued on next page*

*Figure 1 continued*

frequency and amplitude of model neuronal population activity, and in (**D**) at fixed levels of reduction (10%, 20%) in $g_{NaP}$ from control. Color scale bars for values of burst frequency and amplitude are at right of plots in (**A**) and (**C**). Frequencies and amplitudes of population activity in (**C**) and (**D**) are normalized to control values.

The online version of this article includes the following source data and figure supplement(s) for figure 1:

**Source data 1.** Related to *Figure 1A–D*.

**Figure supplement 1.** Comparison of model simulation predictions with a lower synaptic connection probability of 13% ($p_{Syn}$=0.13), as experimentally approximated by *Rekling et al., 2000*, with those presented in *Figure 1*.

**Figure supplement 1—source data 1.** Related to *Figure 1—figure supplement 1A-D*.

change these predictions when the synaptic strength remains constant (see *Figure 1—figure supplement 1*), which allows for a fair assessment of the sensitivity of the network behavior to changes in connection probability (see also *Phillips et al., 2019*). The contrasting predictions for effects of $g_{NaP}$ or $g_{CAN}$ on network dynamics provide a clear basis for experimental testing of the model.

## Experimental design and results

For comparisons with modeling results, we used electrophysiological recording from the preBötC inspiratory population and the hypoglossal (XII) motor output, in conjunction with pharmacological and optogenetic manipulations of preBötC glutamatergic neurons in the in vitro slices. Slices were prepared from transgenic mice that selectively express Channelrhodopsin-2 (ChR2) in glutamatergic neurons by Cre-dependent targeting via the vesicular glutamate transporter type-2 (VgluT2) promoter (*Figure 2*). ChR2-based photostimulation of preBötC glutamatergic neurons bilaterally enabled graded control of the baseline depolarization of this excitatory population to define the frequency tuning curves and associated population bursting amplitudes at different levels of pharmacological block of $g_{NaP}$ or $g_{CAN}$.

## Cre-dependent targeting of preBötC glutamatergic neurons for photostimulation experiments

For optogenetic photostimulation, we established and validated a triple transgenic VgluT2-tdTomato-ChR2-enhanced yellow fluorescent protein (EYFP) mouse line, produced by crossing a previously validated VgluT2-Cre-tdTomato mouse (*Koizumi et al., 2016*) with a Cre-dependent ChR2-EYFP mouse strain. We verified Cre-driven ChR2-EYFP expression in VgluT2-positive tdTomato-labeled neurons within the preBötC region by two-photon microscopy in live neonatal mouse medullary slice (n=6) preparations in vitro, documenting heavy expression of the ChR2-EYFP fusion protein in processes and somal membranes of VgluT2-tdTomato-labeled neurons (*Figure 2A*). We also quantified by whole-cell patch-clamp recordings the photostimulation-induced depolarization of rhythmically active, ChR2-expressing inspiratory glutamatergic neurons (n=12 neurons from four slice preparations from VgluT2-tdTomato-ChR2- EYFP mouse line) at different levels of photostimulation (473 nm of variable laser power ranging from 0.5 to 5 mW) in the rhythmically active in vitro neonatal medullary slice preparations. These in vitro slices effectively isolate the bilateral preBötC along with hypoglossal (XII) motoneurons and nerves to monitor inspiratory XII activity (*Koizumi et al., 2013*; *Koizumi et al., 2016*), allowing laser illumination of the preBötC region unilaterally or bilaterally as well as whole-cell patch-clamp recordings from rhythmically active inspiratory preBötC neurons during laser illumination (*Figure 2B*). We performed targeted current-clamp recordings from rhythmic tdTomato-labeled VgluT2-positive neurons, in which co-expression of ChR2-EYFP fusion protein in the cell membranes was confirmed with two-photon, single optical plane live images (*Figure 2B*, right). Representative examples of ChR2-mediated membrane depolarization (*Figure 2C*) show that the membrane potential of a functionally identified VgluT2-positive preBötC inspiratory neuron was depolarized by ~6.0 mV at 1 mW, ~8.5 mV at 2 mW, and ~11.5 mV at 5 mW laser illumination. The light-induced depolarization had fast kinetics, occurring within ~50 ms and recovery within ~150 ms after terminating illumination. Summary data (n=12 neurons, mean ± SEM, *Figure 2D*) illustrates the laser-power-dependent ChR2-mediated depolarization of VgluT2-positive inspiratory preBötC neurons (4.73±0.24 mV at 0.5 mW, 6.01±0.21 mV at 1 mW, 8.55±0.29 mV at 2 mW, 10.21±0.34 mV at 3 mW, 11.13±0.3 mV at 4 mW, and 11.6±0.35 mV at 5 mW). The membrane depolarizations are

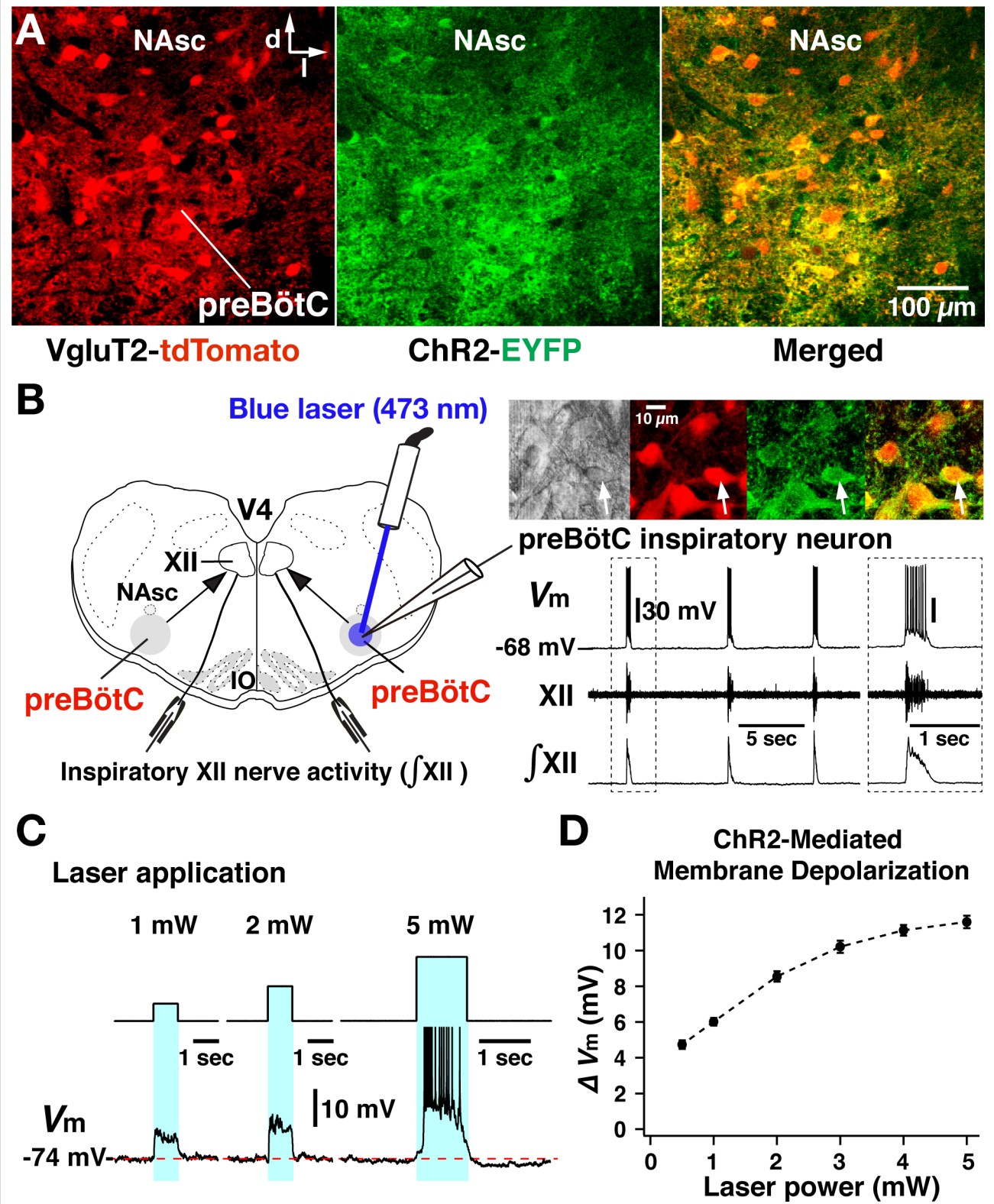

**Figure 2.** Channelrhodopsin-2 (ChR2)-mediated membrane depolarization of preBötzinger complex (preBötC) vesicular glutamate transporter type-2 (VgluT2)-positive inspiratory neurons in vitro. (**A**) Two-photon microscopy single optical plane live images of the preBötC subregion in an in vitro neonatal medullary slice preparation from the VgluT2-tdTomato-ChR2-EYFP transgenic mouse line, illustrating tdTomato-labeled VgluT2-positive neurons distributed in the preBötC and adjacent regions (red) and expression of ChR2-EYFP (green) in somal membranes and neuronal processes of

*Figure 2 continued on next page*

*Figure 2 continued*

VgluT2-tdTomato neurons, as seen in the merged image. Abbreviations: d, dorsal; l, lateral; NAsc, semi-compact subdivision of nucleus ambiguus. (**B**) Overview of experimental in vitro rhythmic slice preparation from a neonatal VgluT2-tdTomato-ChR2 transgenic mouse showing whole-cell patch-clamp recording from a functionally identified preBötC inspiratory VgluT2-positive neuron with unilateral preBötC laser illumination (0.5–5 mW) to test for neuronal expression of ChR2, and suction-electrode extracellular recordings from hypoglossal (XII) nerves to monitor inspiratory activity. Regional photostimulation was accomplished with a 100-µm diameter optical cannula positioned at the slice surface above the preBötC region. Abbreviations: V4, fourth ventricle; IO, inferior olivary nucleus. Upper right in (**B**) two-photon single optical plane images showing an imaged preBötC inspiratory neuron (arrow) targeted for whole-cell recording. From left, Dodt gradient contrast structural image, VgluT2-Cre driven tdTomato labeling, ChR2-EYFP expression, and merged image that confirm co-expression of tdTomato and ChR2-EYFP. The lower right traces are current-clamp recordings from the VgluT2-positive preBötC neuron shown in the above images, illustrating inspiratory spikes/bursts synchronized with integrated inspiratory XII nerve activity (∫XII). The traces shown at right within the dashed box are expanded time scale traces of the bursting activity within the dashed box in the left traces. (**C**) The membrane potential (*Vm*) of the neuron shown in (**B**) was depolarized during photostimulation by ~6.0 mV at 1 mW, ~8.5 mV at 2 mW, and ~11.5 mV at 5 mW of laser power (spikes are truncated). The neuron was hyperpolarized from resting baseline potential to –74 mV by applying constant current in this example to reveal the magnitude of the light-induced membrane depolarization. Photostimulation was performed in the interval between inspiratory population bursts. (**D**) Summary data (n=12 neurons from four slice preparations, mean ± SEM) showing the laser-power-dependent, ChR2-mediated neuronal membrane depolarization from baseline (ΔV$_M$) of VgluT2-positive preBötC inspiratory neurons.

The online version of this article includes the following source data for figure 2:

**Source data 1.** Related to *Figure 2D*.

significantly different (p<0.01 by ANOVA test comparing these groups). The subsequent Tukey's HSD (honestly significant difference) test for pairwise comparison shows a significant difference with p<0.01 in all cases except p=0.4542 for 4 mW vs 5 mW. These results enabled us to define the relation between laser power and ChR2-mediated membrane depolarization for the inspiratory glutamatergic neuron population.

## Perturbations of inspiratory rhythm and burst amplitude by photostimulation of glutamatergic neurons within the preBötC region

We performed site-specific bilateral photostimulations of the preBötC VgluT2-positive neuron population (*Figure 3A*) by sustained laser illumination (20 ms pulses at 20 Hz, 473 nm, 30–90-s pulse train durations) at graded laser powers between 0.25 and 2.0 mW (n=31 slice preparations). To systematically analyze relations between inspiratory burst frequency/burst amplitude and laser power, we applied single epochs of laser illumination and allowed for recovery from each epoch before changing the laser power (representative examples with simultaneous population activity recordings from the preBötC and XII are shown in *Figure 3B*, and summary data [mean ± SEM] is shown in *Figure 3C*). Bilateral photostimulation (0.25–2 mW) of the preBötC caused rapid and reversible increases of inspiratory burst frequency in a laser-power-dependent manner (85±7% increase at 0.25 mW, 174±7% at 0.5 mW, 262±6% at 1 mW, 326±6% at 2 mW, Wilcoxon matched-pairs signed rank test, p<0.01 in all cases). After termination of laser illumination, an inhibition of rhythmicity followed and subsequently recovered to the control frequency. The average recovery time (seconds) at each laser power was 48±4 at 0.25 mW, 88±4 at 0.5 mW, 128±9 at 1 mW, 182±11 at 2 mW, respectively. Concurrently, there was a significant laser power-dependent decrease of burst amplitude of inspiratory preBötC population and inspiratory XII activity during photostimulation (8±2% decrease at 0.25 mW, 15±2% at 0.5 mW, 25±3% at 1 mW, 44±4% at 2 mW, p<0.01 in all cases by Wilcoxon signed rank test). Bilateral photostimulation also induced tonic activity in preBötC population recordings, which was of higher amplitude at higher laser power, but did not consistently induce tonic activities in XII nerve output. Otherwise, XII burst amplitude and frequency always directionally mirrored preBötC glutamatergic population activity.

These results demonstrate that preBötC glutamatergic neurons play an important role in regulating the inspiratory rhythm with a membrane voltage-dependent frequency control mechanism in the neonatal medullary slices in vitro. We note that in control experiments with rhythmic slice preparations from the VgluT2-tdTomato, non-ChR2-expressing transgenic mice (n=6), we tested for photo-induced perturbations of inspiratory burst frequency and amplitude, and confirmed that there were no significant perturbations of frequency and amplitude as a function of laser power (0.25–2 mW; ANOVA tests for frequency and burst amplitude, p=0.545 and 0.761, respectively).

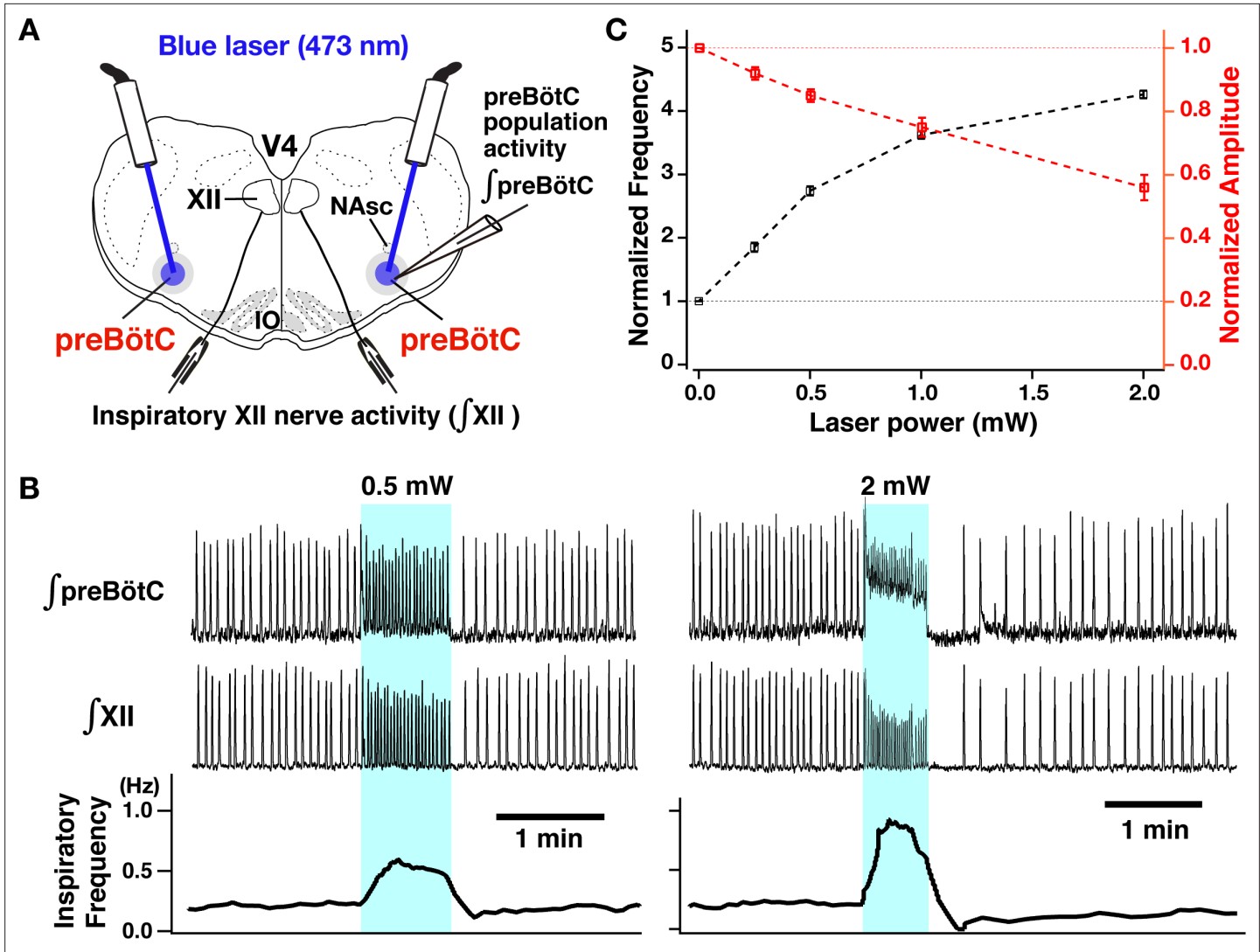

**Figure 3.** Photostimulation of the bilateral preBötzinger complex (preBötC) vesicular glutamate transporter type-2 (VgluT2)-positive neuron population caused laser-power-dependent increases of inspiratory burst frequency and decreases of burst amplitude. (**A**) Overview of experimental in vitro rhythmically active slice preparation from neonatal VgluT2-tdTomato-channelrhodopsin-2 transgenic mouse with macro-patch electrodes on the preBötC region for recording preBötC population activity and suction electrodes on hypoglossal (XII) nerves to monitor inspiratory motor activity during bilateral preBötC laser illumination (0.25–2 mW, 473 nm wavelength) with optical cannula (100 µm diameter) positioned obliquely to illuminate the preBötC region. Abbreviations: NAsc, nucleus ambiguus semi-compact subdivision; V4, fourth ventricle; IO, inferior olivary nucleus. (**B**) Representative examples of epochs of bilateral preBötC laser illumination and effects on inspiratory burst frequency and burst amplitude. The upper traces show the integrated macro-patch recordings from the preBötC inspiratory neuron population (∫preBötC), middle traces show the integrated inspiratory XII activity (∫XII), and the bottom traces show the inspiratory burst frequency (time-based moving median in a 10 s window). The sustained laser illumination with the laser intensity is given by blue shading. Low-intensity illumination (0.5 mW) caused significant increase (~149% in this example) of inspiratory burst frequency and decrease of inspiratory burst amplitude of both preBötC and XII population activity (~22 and~21%, respectively, left panel). Higher intensity laser illumination (2 mW; right panel) caused a larger increase (~295%) of inspiratory frequency and decrease of burst amplitude of inspiratory preBötC and XII activity (~65 and~44%, respectively). Also note that photostimulation induced tonic activity (indicated by baseline shift) in ∫preBötC population activity recordings. (**C**) Summary data of relations between inspiratory burst frequency and amplitude vs photostimulation laser power (n=31 slices; data points plotted are mean ± SEM).

The online version of this article includes the following source data for figure 3:

**Source data 1.** Related to *Figure 3C*.

# Regulation of burst frequency and amplitude by I$_{NaP}$ in preBötC glutamatergic neurons analyzed with combined optogenetic and pharmacological manipulations

After defining the voltage-dependent regulation of burst frequency and amplitude as described above, we investigated how pharmacological block of I$_{NaP}$ in preBötC glutamatergic neurons affects this voltage-dependent control in the rhythmically active in vitro medullary slice preparations from the VgluT2-tdTomato-ChR2 mouse line. We first analyzed the pharmacological profile of block of I$_{NaP}$ in preBötC inspiratory glutamatergic neurons with the I$_{NaP}$ blockers tetrodotoxin citrate (TTX) at very low concentrations (5–20 nM, n=8) and also riluzole (5–20 µM, n=6) bath applied in the neonatal mouse medullary slice preparations as we have previously done for neonatal rat preBötC inspiratory neurons (*Koizumi and Smith, 2008*), but now analyzed for genetically identified mouse glutamatergic neurons. Whole-cell voltage-clamp recording from optically identified VgluT2-tdTomato-expressing inspiratory neurons was used to obtain current-voltage (I-V) relationships by applying slow voltage ramps (30 mV/s; –100 to +10 mV) and we computed TTX- and riluzole-sensitive I$_{NaP}$ by subtracting I-V curves obtained before and after block of I$_{NaP}$ with TTX or riluzole (*Figure 4—figure supplement 1*). Both TTX and riluzole attenuated the peak I$_{NaP}$ amplitude (measured at ~–40 mV) in a dose-dependent manner. TTX reduced peak I$_{NaP}$ by 36–63% at 5 nM, by 74–94% at 10 nM, and completely blocks I$_{NaP}$ at 20 nM. Riluzole reduced the peak I$_{NaP}$ by 36–66% at 5 µM, by 82–95% at 10 µM, and completely blocked I$_{NaP}$ at 20 µM. In this dataset, there were five endogenous burster inspiratory neurons, which had significantly higher g$_{NaP}$/Cm (105.9±6.5 pS/pF) compared to non-endogenous burster inspiratory neurons (49.6±3.5 pS/pF) (p<0.05 by non-parametric Kolmogorov-Smirnov test), comparable to results from a previous study of neuronal properties performed in neonatal rat slices in vitro (*Koizumi and Smith, 2008*). However, the I$_{NaP}$ attenuation by the blockers was not significantly different for these two groups (TTX: p=0.464 at 5 nM, p=0.982 at 10 nM, p=0.857 at 20 nM; riluzole: p=0.933 at 5 µM, p=0.4 at 10 µM, p=0.933 at 20 µM by non-parametric Kolmogorov-Smirnov tests). These pharmacological profiles are also comparable to the data obtained in neonatal rat preBötC inspiratory neurons in our previous in vitro studies (*Koizumi and Smith, 2008*). We note that these pharmacological profiles were obtained for neurons recorded at depths up to 50–75 µm in the slices; neurons located deeper in the slice tissue may exhibit less reduction of I$_{NaP}$ at the time points when these more superficial neurons were analyzed due to tissue penetration-related, spatiotemporal non-uniformity of I$_{NaP}$ attenuation.

We then comparatively analyzed control of XII inspiratory burst frequency and amplitude by bilateral preBötC photostimulation (473 nm, 20 Hz pulses, 30–90 s, trains of sustained photostimulation as described above) under I$_{NaP}$ pharmacological blockade. As shown in *Figure 4A and B*, partial block of I$_{NaP}$ (5–10 nM TTX, or 5–10 µM riluzole) gradually reduced XII inspiratory burst frequency, slightly reduced amplitude, and terminated inspiratory rhythmic bursting in almost all cases (n=6/7 at 5 nM TTX, n=8/8 at 10 nM TTX, n=6/7 at 5 µM riluzole, n=8/8 at 10 µM riluzole). In the cases where rhythmic bursting was terminated, after 30 min, we again bilaterally photostimulated the preBötC, which restored rhythmic bursting with voltage-dependent frequency control (0.25–2.0 mW) in all cases (n=14/14 with TTX, n=14/14 with riluzole). The summary data (n=6 slice preparations for TTX 5 nM, n=8 for TTX 10 nM, n=6 for riluzole 5 µM, n=8 for riluzole 10 µM, mean ± SEM plotted) in *Figure 4C* (left) shows that the inspiratory burst frequency significantly increases as a function of laser power (Wilcoxon signed rank test, p<0.05 in all cases). The frequency tuning curves (control, TTX 5 nM, TTX 10 nM, riluzole 5 µM, and riluzole 10 µM) are significantly different by ANOVA test (p<0.01). The tuning curves with TTX and riluzole shifted downward relative to control (TTX 5 nM vs control, TTX 10 nM vs control, riluzole 5 µM vs control, riluzole 10 µM vs control; in all cases, p<0.01 by subsequent Tukey's HSD test), and more downwardly shifted with higher drug concentrations (Tukey's HSD test, TTX 5 nM vs TTX 10 nM, p<0.01; riluzole 5 µM vs riluzole 10 µM, p<0.01). The frequency-laser power relations are not significantly different for the TTX and riluzole groups (Tukey's HSD test, p=0.927 for TTX 5 nM vs riluzole 5 µM; p=0.969 for TTX 10 nM vs riluzole 10 µM).

Summary data (n=6 slice preparations at TTX 5 nM, n=8 at TTX 10 nM, n=6 at riluzole 5 µM, n=8 at riluzole 10 µM, mean ± SEM plotted) of the relations between normalized inspiratory burst amplitude and laser power presented in *Figure 4C* (right) show a significant laser-power-dependent decrease of inspiratory burst amplitude with I$_{NaP}$ attenuation (Wilcoxon signed rank test, p<0.05 in all cases). Although significant, changes in amplitude relative to control are smaller with TTX block of I$_{NaP}$. The

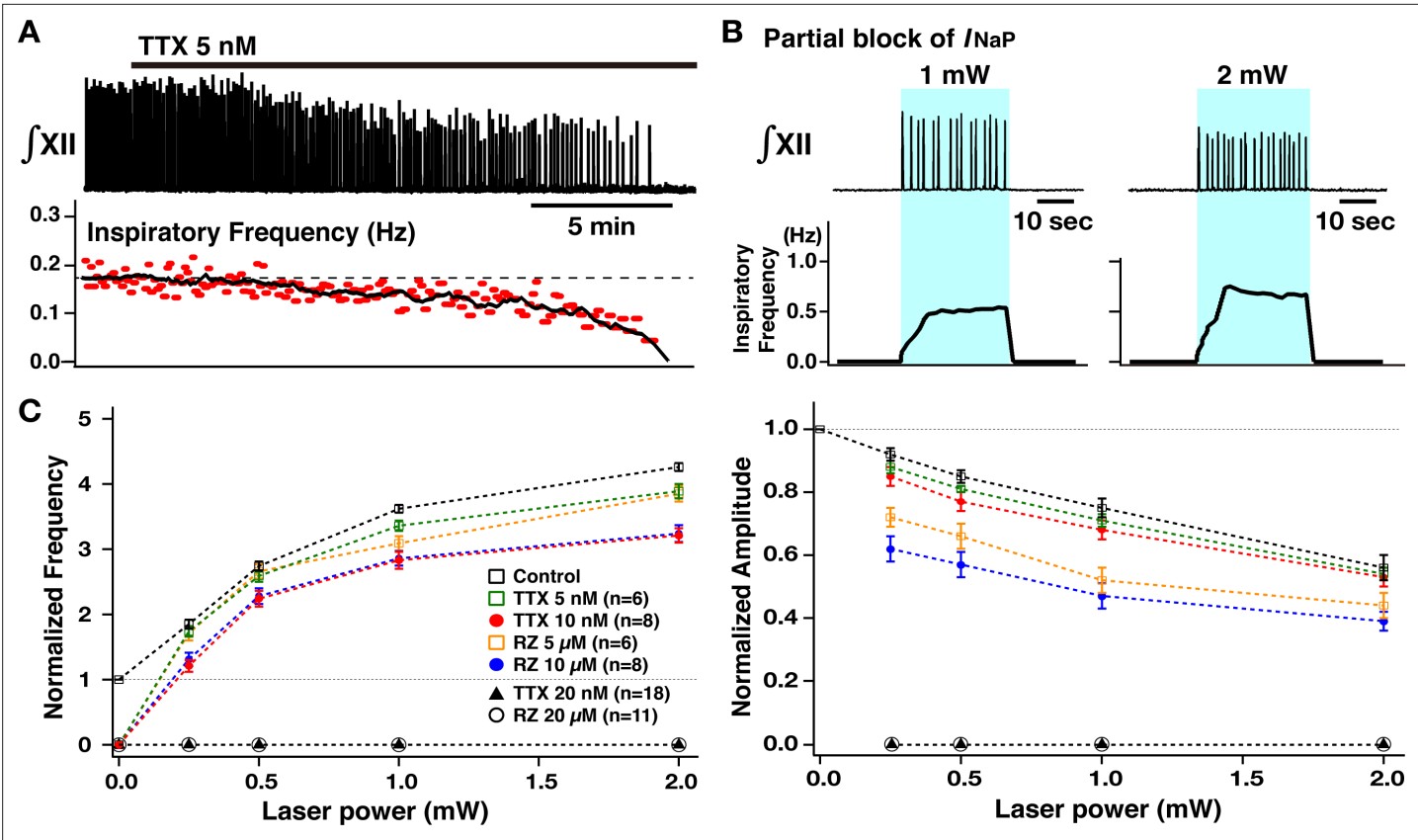

**Figure 4.** Perturbations of inspiratory burst frequency and amplitude by bilateral preBötzinger complex (preBötC) photostimulation during pharmacological block of neuronal persistent sodium current ($I_{NaP}$). (**A**) Example recordings of integrated XII activity (∫XII) with bath application of low concentration of tetrodotoxin citrate (TTX; 5 nM), which gradually decreased inspiratory burst frequency and completely stopped the rhythm within ~25–30 min. The amplitude of XII activity also gradually decreased (~36% before the rhythm stopped in this example). (**B**) Under these conditions of partial block of $I_{NaP}$, bilateral preBötC photostimulation (30 min after the rhythm stopped, blue-shaded epoch) could reinitiate the rhythm, which was also laser-power-dependent (~202% increase at 1 mW compared to the control, and ~306% increase at 2 mW in the example shown). (**C**) Summary plots (TTX 5 nM, n=6; TTX 10 nM, n=8; TTX 20 nM, n=18; riluzole [RZ] 5 μM, n=6; RZ 10 μM, n=8; RZ 20 μM, n=8; mean ± SEM plotted) of the relations between laser power and normalized inspiratory burst frequency indicate laser-power-dependent, significant increases of frequency in all cases. The curves for higher concentration of TTX and riluzole are downward-shifted compared to those for the lower concentration as well as those under control conditions (before drug applications). Right panel shows summary plots (TTX 5 nM, n=6; TTX 10 nM, n=8; TTX 20 nM, n=18; RZ 5 μM, n=6; RZ 10 μM; n=8; RZ 20 μM, n=8; mean ± SEM plotted) of the relations between laser power and normalized XII burst amplitude indicating laser-power-dependent, significant decreases of burst amplitude in all cases. The curves under the $I_{NaP}$ partial block (except for at TTX 5 nM) are downward-shifted compared to those under the control conditions. The decrease in burst amplitudes under RZ is more significant than those under TTX, although there are no significant differences between different concentrations of TTX or RZ. The loss of network rhythmic bursting activity after complete block of $I_{NaP}$ (20 nM TTX or 20 μM RZ) is reflected by the zero frequency (left) and amplitude (right) points for 20 μM RZ and 20 nM TTX on the plots.

The online version of this article includes the following source data and figure supplement(s) for figure 4:

**Source data 1.** Related to *Figure 4C*.

**Figure supplement 1.** Pharmacological profile of block of neuronal persistent sodium current ($I_{NaP}$) in preBötzinger complex (preBötC) inspiratory glutamatergic neurons.

**Figure supplement 1—source data 1.** Related to *Figure 4—figure supplement 1C*.

amplitude vs laser power relation curves with partial $I_{NaP}$ blockade are significantly different (p<0.01 by ANOVA). The curves with $I_{NaP}$ attenuation (except for TTX 5 nM) are shifted significantly downward compared to control conditions (Tukey's HSD test, TTX 5 nM vs control, p=0.304; TTX 10 nM vs control; riluzole 5 μM vs control, riluzole 10 μM vs control, p<0.05), but there were no significant differences for the relations with different concentrations of TTX or riluzole (TTX 5 nM vs TTX 10 nM, p=0.986; riluzole 5 μM vs riluzole 10 μM, p=0.259 by Tukey's HSD test). The curves obtained under

riluzole were more significantly downward-shifted than those under TTX (TTX 5 nM vs riluzole 5 µM, p<0.01; TTX 10 nM vs riluzole 10 µM, p<0.01 by Tukey's HSD test).

Under conditions of complete block of $I_{NaP}$ (>30 min after either 20 nM TTX or 20 µM riluzole), bilateral sustained preBötC photostimulation could not reinitiate the inspiratory rhythm at any laser power ranging from 0.25 to 5 mW (n=18/18 under TTX, n=11/11 under riluzole; *Figures 4 and 5*), but induced only graded tonic spiking in preBötC neurons with progressive increases of laser power >1 mW after block of $I_{NaP}$. This tonic activity without rhythmic bursting was also verified at the preBötC excitatory population activity level (*Figure 5*).

We performed power spectrum analyses to determine if there are rhythmogenic mechanisms inherent within the preBötC excitatory network that do not depend on $I_{NaP}$ and emerge with the graded levels of tonic excitatory population activity at the various levels of sustained bilateral photostimulation. These spectral analyses of the recorded graded tonic activity after block of $I_{NaP}$ indicated that there was no rhythmic activity at each level of the photostimulation-induced tonic activity in comparison to the control activity. In control conditions there is clear definition of spectral peaks for the fundamental and higher harmonic frequencies as given by the Fast Fourier Transform of the rectified preBötC activity signal. This power spectrum reflects the periodicity and (non-sinusoidal) shape of the bursts of population activity, but the spectrum is flat after block of $I_{NaP}$ (*Figure 5*, n=5), reflecting a complete absence of rhythmic activity. At the cellular level, current-clamp and voltage-clamp recordings confirmed the absence of neuronal rhythmic synaptic drive potentials/currents in preBötC inspiratory glutamatergic neurons (*Figure 6*).

These results demonstrate that $I_{NaP}$ in preBötC glutamatergic neurons is critically involved in generating the inspiratory rhythm with a voltage-dependent frequency control mechanism in vitro. The results also indicate that there are no apparent rhythmogenic mechanisms inherent within the excitatory network that can emerge at various levels of excitation after eliminating the $I_{NaP}$-dependent mechanisms.

## Regulation of inspiratory burst frequency and amplitude by $I_{CAN}$/TRPM4 in preBötC glutamatergic neurons

We next investigated contributions of $I_{CAN}$/TRPM4 to the voltage-dependent control of XII inspiratory burst frequency and amplitude for comparisons to model predictions. We analyzed perturbations of burst frequency and amplitude by bilateral preBötC photostimulation (as described above) after pharmacologically blocking TRPM4 with a specific inhibitor of TRPM4 channels (9-phenanthrol, a putative blocker of $I_{CAN}$). Bath application of 9-phenanthrol at a concentration proposed to be selective for TRPM4 (50 µM) to the rhythmically active slice preparations from VgluT2-tdTomato-ChR2 neonatal mice (*Figure 7A*) significantly reduced the amplitude of inspiratory XII bursts (43±4% reduction, n=8, p<0.01 by Wilcoxon signed rank test), with little effects on burst frequency (1±1% increase, n=8, Wilcoxon signed rank test, p=0.461) in these slices, as we have previously described (*Koizumi et al., 2018*). Under these TRPM4 block conditions (steady-state at >60 min after 9-phenanthrol application, *Figure 7C*), photostimulation caused a significant increase of inspiratory burst frequency compared to control conditions at a given laser power (0.25–2.0 mW, *Figure 7D* black lines, Wilcoxon signed rank test, p<0.01 in all cases), accompanied by a significant laser-power-dependent decrease of burst amplitude compared to control (*Figure 7D* red lines, Wilcoxon signed rank test, p<0.01 in all cases). The relations between normalized inspiratory burst frequency and amplitude vs laser power summarized in *Figure 7D* (n=8 slice preparations, mean ± SEM plotted) show that TRPM4 block significantly shifts the frequency tuning curve upward compared to control conditions (black lines in *Figure 7D*, Wilcoxon signed rank test, p<0.01). The relation between burst amplitude and laser power is shifted downward significantly compared to control conditions (red lines in *Figure 7D*, Wilcoxon signed rank test, p<0.01). Note that under baseline conditions of excitability, TRPM4 block does not alter burst frequency, and the upward shift in the frequency tuning curve is opposite to the shift seen with TTX or riluzole block of $I_{NaP}$. These findings are consistent with the model predictions presented in *Figure 1*.

## Model simulations-data comparisons

For comparisons of model predictions and results from the experimental photostimulation and pharmacological manipulations (*Figure 8*), we incorporated in the original model (*Phillips et al., 2019*) a ChR2 current ($I_{ChR2}$) to represent $I_{App}$ for the excitatory neuron population and mimic effects of

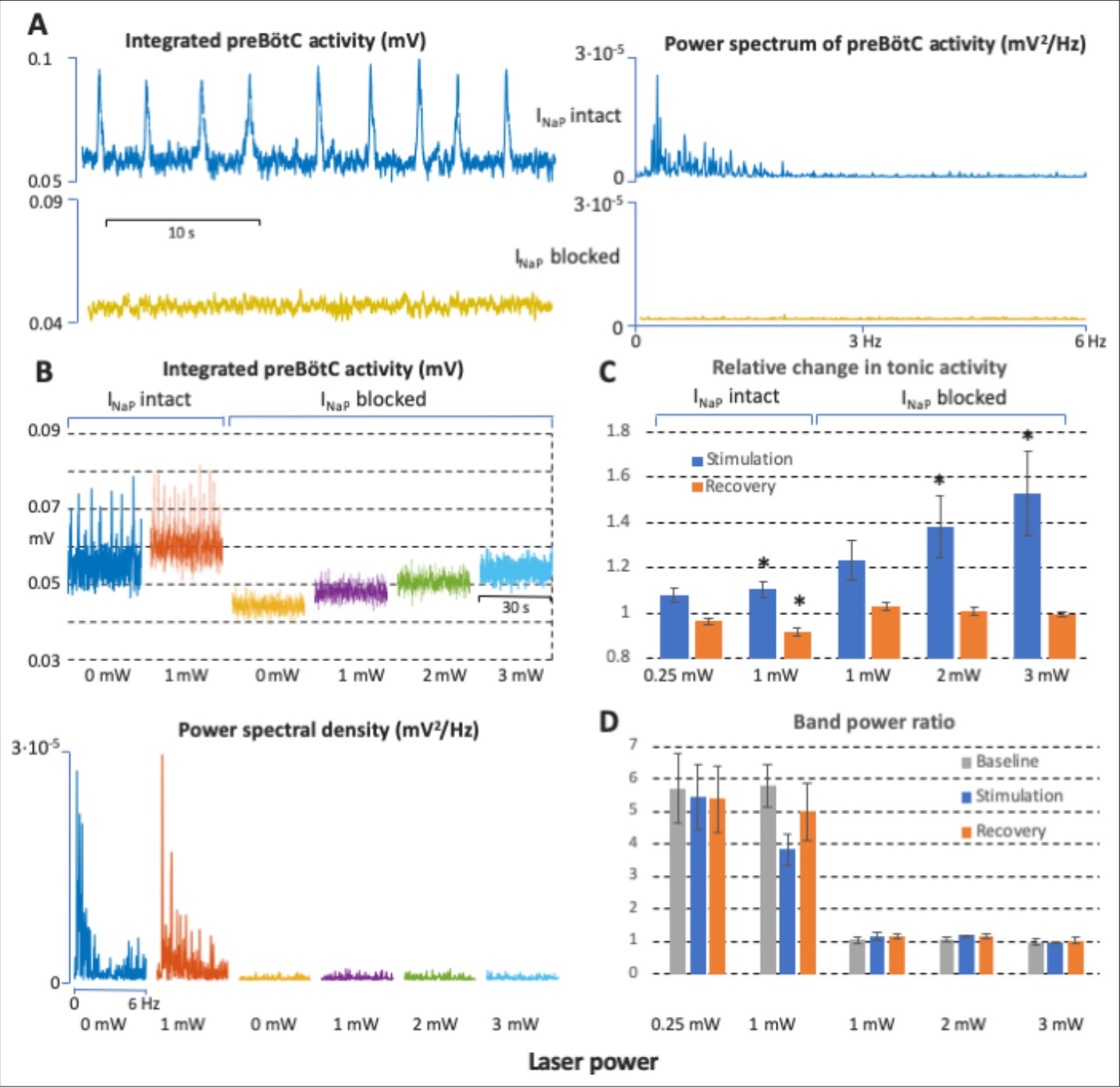

**Figure 5.** Power spectrum analyses of preBötzinger complex (preBötC) neuronal population activity in vitro before and after complete block of neuronal persistent sodium current ($I_{NaP}$). (**A**) Examples of preBötC integrated population activity patterns and associated power spectra before and after block of $I_{NaP}$, which eliminates rhythmic preBötC population activity, leaving only a baseline low level of tonic activity or 'noise' with a flat power spectrum (lower traces). The power spectra of the rhythmic activity have clear peaks corresponding to the fundamental and higher harmonic frequencies (upper traces). (**B**) Representative experiment illustrating activity patterns and power spectra before and after block of $I_{NaP}$ at various levels of photostimulation (0 and 1 mW in control; 0, 1, 2, and 3 mW after $I_{NaP}$ blocked). With $I_{NaP}$ intact (control conditions, left traces), photostimulation increases the frequency of integrated preBötC inspiratory population activity accompanying an upward shift of the integrated baseline activity due to tonic activity, and there are clear peaks in the power spectra of the rhythmic activity corresponding to the fundamental and higher harmonic frequencies (lower panel). After block of $I_{NaP}$ with 20 nM tetrodotoxin citrate (TTX), there is a graded shift in the baseline level of tonic activity during photostimulation, but no rhythmic integrated population activity as indicated by the flat power spectra resembling that of the baseline noise activity in the absence of photostimulation. (**C**) Data from integrated preBötC population recordings showing the change in level of tonic activity relative to baseline during graded

*Figure 5 continued on next page*

*Figure 5 continued*

photostimulation and the recovery period following stimulation under control conditions with $I_{NaP}$ intact and with $I_{NaP}$ blocked (20 nM TTX). Increasing the level of photostimulation (0.25, 1 mW in control; 1, 2, 3 mW with $I_{NaP}$ blocked) in both conditions increases the level of tonic activity. The activity returns to the baseline level ($I_{NaP}$ blocked) or is slightly depressed (control conditions) in the post-photostimulation recovery period. Bars indicate mean ± SEM from 5 slices. * indicates statistically significant difference from unity (p<0.05 by two-tailed t-test). (**D**) Quantification of the oscillatory component in the preBötC activity before and after complete block of $I_{NaP}$ at different levels of photostimulation as the ratio of the band power in the 0–3 Hz frequency range over the band power in the 3–6 Hz range. The ratios are high in control conditions, reflecting significantly higher spectral power content in the low frequency band (oscillations) compared to one in the high frequency band (noise) when there is rhythmic activity at baseline, with photostimulation, and during recovery. The ratios are not different and have a value of unity when $I_{NaP}$ is blocked, reflecting the flat power spectra under this condition. Bars indicate mean ± SEM from 5 slices.

The online version of this article includes the following source data for figure 5:

**Source data 1.** Related to *Figure 5C and D*.

photo-induced neuronal depolarization. ChR2 was modeled by a four-state Markov channel (see *Figure 9*), based on biophysical representations of this channel in the literature (*Williams et al., 2013*), which we found could be parameterized as described in Materials and methods to closely match our experimental data on relations between laser power and membrane depolarization of preBötC

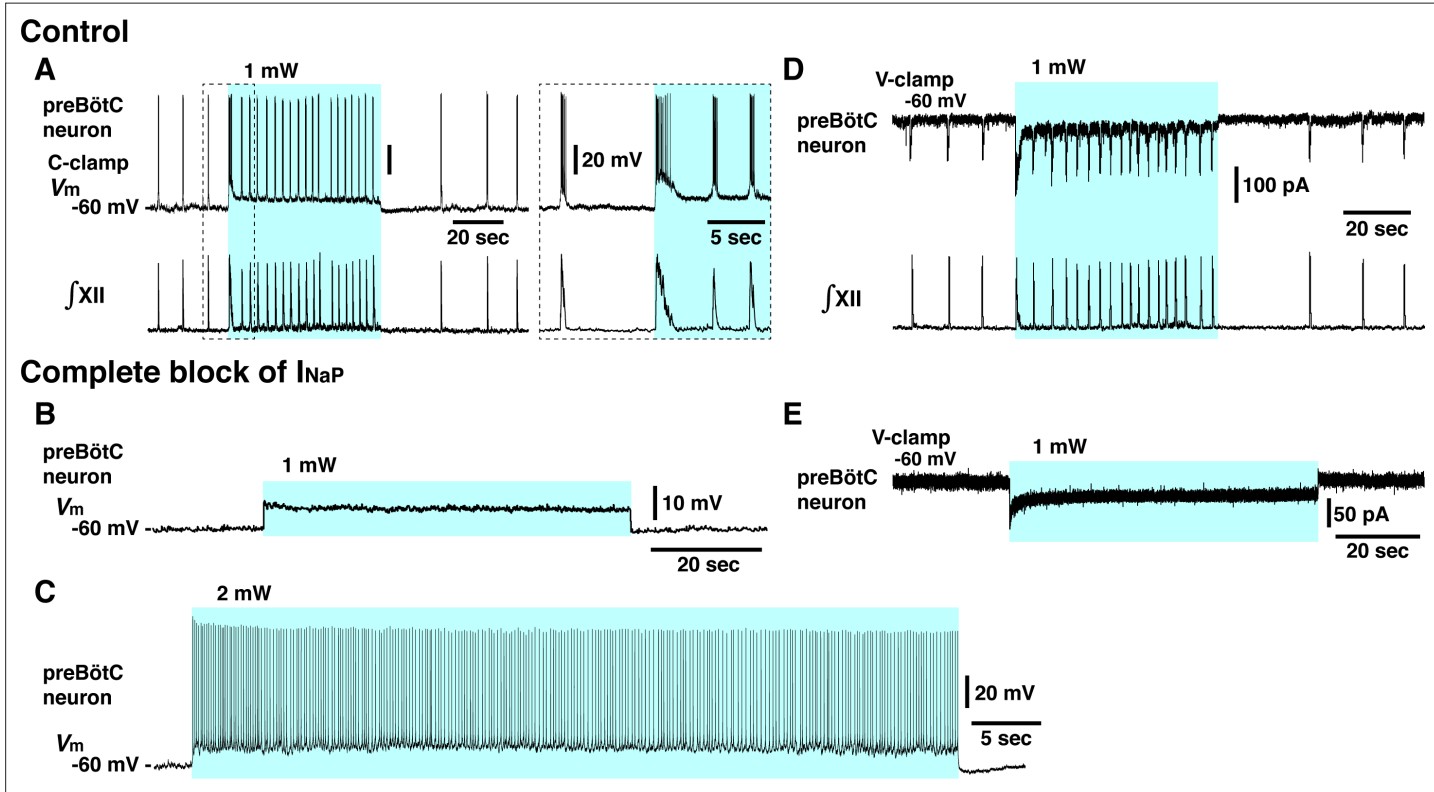

**Figure 6.** Elimination of inspiratory rhythm at the cellular level after complete block of neuronal persistent sodium current ($I_{NaP}$) in vitro. (**A**) Whole-cell current-clamp recordings from td-tomato-labeled preBötzinger complex (preBötC) inspiratory neuron illustrating rhythmic bursting synchronized with inspiratory hypoglossal (XII) motor activity. In control conditions, optogenetic stimulation (1 mW) induced neuronal membrane depolarization (~6 mV) along with a significant increase of inspiratory bursting frequency synchronized with the network bursting frequency indicated by the integrated inspiratory XII activity (∫XII). Shown at right within the dashed box are expanded time scale traces of the activity within the dashed box in the left traces. (**B, C**) With complete block of $I_{NaP}$ (20 nM tetrodotoxin citrate [TTX]), photostimulation did not induce rhythmic activity in this neuron, only membrane depolarization without rhythmic synaptic drive potentials or bursting at 1 mW laser application (**B**) and only tonic neuronal spiking at a higher laser power of 2 mW (**C**). The latter indicates that the neuron retained spiking capabilities at the low concentration of TTX employed, which did not interfere with action potential generation by transient Na⁺ channels while $I_{NaP}$ was completely blocked (see *Figure 4—figure supplement 1*). (**D**) Voltage-clamp recordings from td-tomato-labeled preBötC inspiratory neuron showing inward rhythmic synaptic drive currents synchronized with inspiratory hypoglossal activities. Under control conditions, optogenetic stimulation (1 mW) induced inward currents and rhythmic synaptic drive currents synchronized with the higher frequency inspiratory XII activities. (**E**) Rhythmic inspiratory synaptic drive currents did not occur with photostimulation (1 mW) after complete block of $I_{NaP}$ at 20 nM TTX, indicating loss of synaptic interactions and network rhythmic activity.

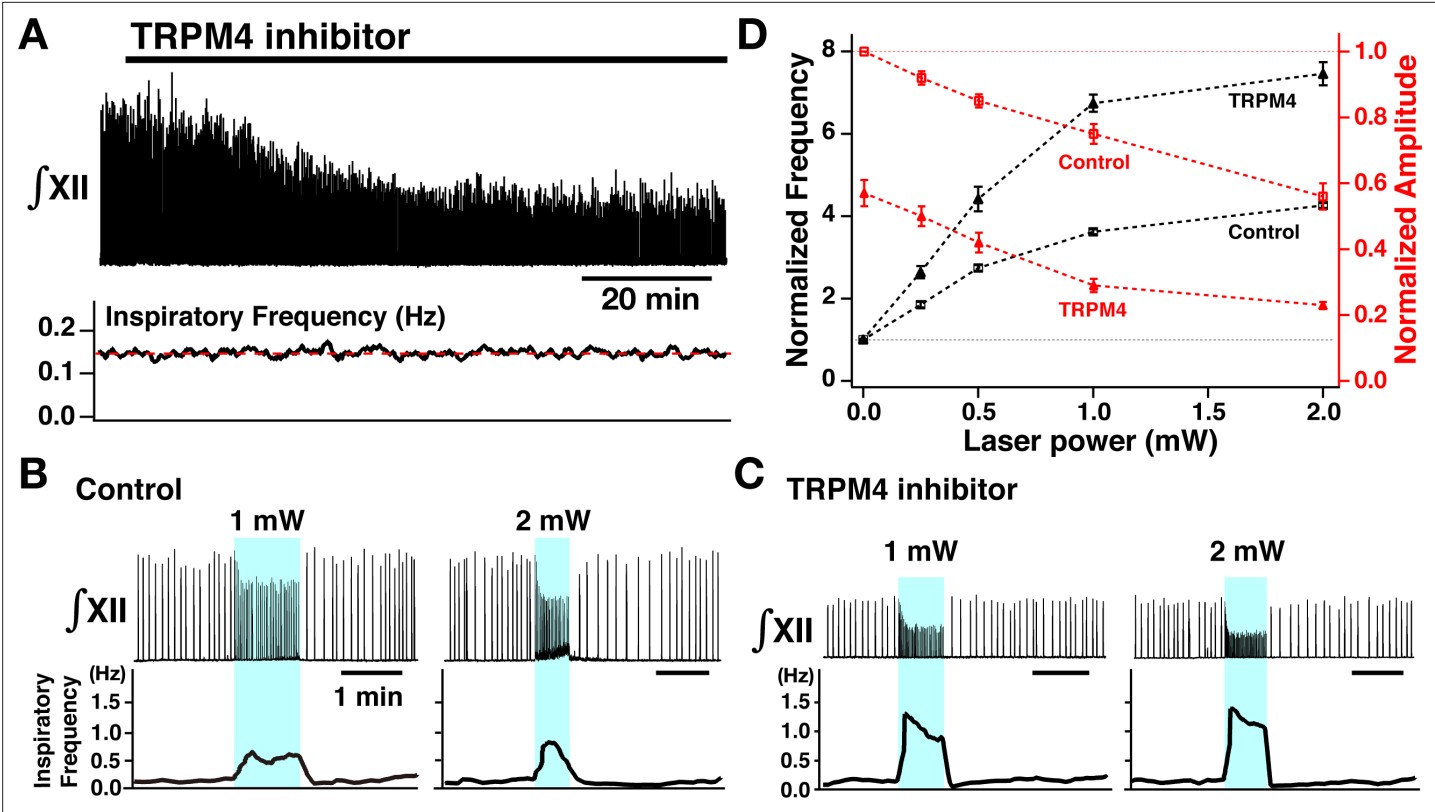

**Figure 7.** Perturbations of inspiratory burst frequency and amplitude by bilateral photostimulation of the preBötzinger complex (preBötC) during transient receptor potential channel M4 (TRPM4) pharmacological blockade. (**A**) Upper trace illustrates the time course of integrated XII inspiratory activity (∫XII) during bath application of the specific pharmacological inhibitor of TRPM4 channels (9-phenanthrol, 50 μM), which gradually decreased inspiratory burst amplitude (~58% reduction at steady state ~60 min after drug application), but had little effect on the inspiratory frequency (bottom trace). (**B**) Bilateral photostimulation of the preBötC (control, before pharmacological block) increased inspiratory burst frequency in a laser-power-dependent manner (~284% increase at 1 mW, ~354% increase at 2 mW) and monotonically decreased inspiratory XII burst amplitude (~25% decrease at 1 mW, ~44% decrease at 2 mW). (**C**) Under TRPM4 block conditions (>60 min after 9-phenanthrol application), photostimulation significantly increased inspiratory frequency (~586% increase at 1 mW, ~724% increase at 2 mW), and decreased ∫XII inspiratory burst amplitude (~72% decrease at 1 mW, ~78% decrease at 2 mW). (**D**) Summary plots of monotonic relations between laser power and normalized inspiratory frequency (black lines, n=8 slices) and normalized amplitude (red, n=8). Data points plotted are mean ± SEM.

The online version of this article includes the following source data for figure 7:

**Source data 1.** Related to *Figure 7D*.

inspiratory glutamatergic neurons (*Figure 8A*). To simulate TTX and riluzole block of $I_{NaP}$, we incorporated their distinct pharmacological mechanisms of action, as well as off-target effects of riluzole. TTX directly obstructs the Na⁺ pore (*Hille, 2001*), whereas riluzole shifts $I_{NaP}$ inactivation dynamics in the hyperpolarizing direction (*Hebert et al., 1994*; *Song et al., 1997*; *Ptak et al., 2005*). Additionally, at the concentrations used for $I_{NaP}$ block in the preBötC (≤20 μM), riluzole attenuates excitatory synaptic transmission (*MacIver et al., 1996*; *Bellingham, 2011*). Therefore, as in previous work (*Phillips and Rubin, 2019*), TTX application was simulated by a direct reduction in $g_{NaP}$, while a hyperpolarizing shift in $V_{h1/2}$ and a reduction in the conductance of recurrent excitatory synaptic connections by 20 and 25% were used to simulate 5 μM and 10 μM riluzole application, respectively. In agreement with the experimental results (*Figure 4*), the model simulations show that TTX or riluzole block of $I_{NaP}$ decreases the population-level bursting frequency and amplitude at each level of laser power (*Figure 8B–C*). This reduction also alters the voltage-dependent rhythmogenic behavior, with a downward shift in the frequency tuning curve such that the frequency dynamic range is reduced and there is a complete termination of rhythm generation at a sufficient level of $I_{NaP}$ block under baseline conditions of network excitability (i.e. laser power = 0 mW). Due to the simulated off-target attenuating effects of riluzole on excitatory synaptic transmission, the downward shift in the amplitude vs laser

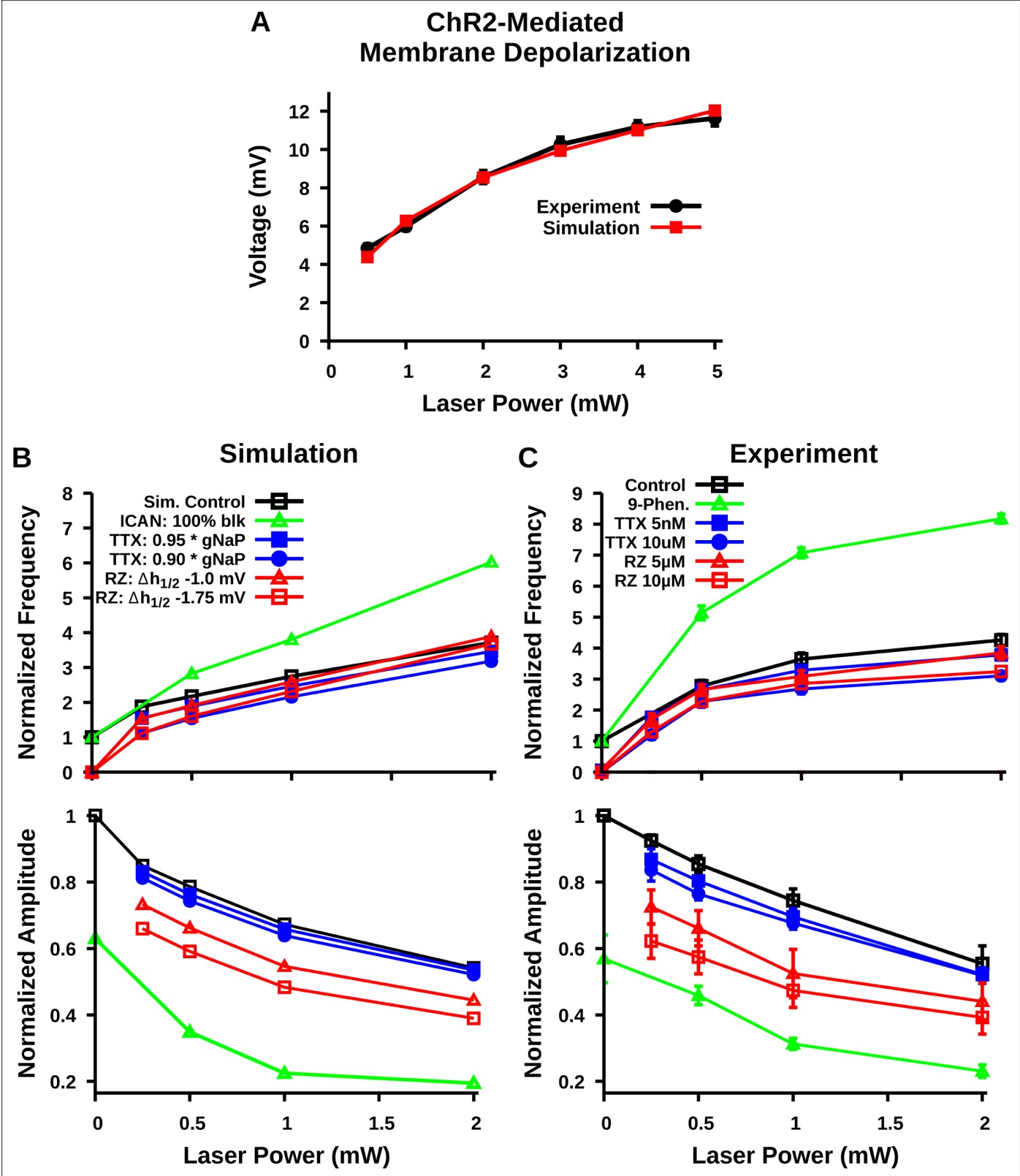

**Figure 8.** Comparison of experimental and simulated optogenetic photostimulation of the preBötzinger complex (preBötC) excitatory network under control conditions and after partial block of neuronal persistent sodium current ($I_{NaP}$) or transient receptor potential channel M4 (TRPM4)/calcium-activated non-selective cation current ($I_{CAN}$). (**A**) Matched relationship between neuronal membrane depolarization from baseline ($\Delta V_M$) as a function of laser power with photostimulation from experimental data (same as shown in **Figure 2D**) and model simulations. (**B**) Relationship between laser power

*Figure 8 continued on next page*

*Figure 8 continued*

and burst frequency (top) and amplitude (bottom) of the preBötC network with model simulated blockade of $I_{NaP}$ by tetrodotoxin citrate (TTX) and riluzole (RZ) or block of $I_{CAN}$. $I_{NaP}$ block by TTX and $I_{CAN}$ block by 9-phenanthrol (Phen.) were simulated by reducing the persistent sodium conductance ($g_{NaP}$) and calcium-activated non-selective cation conductance ($g_{CAN}$), respectively. In contrast, blockade of $I_{NaP}$ by RZ was simulated by a hyperpolarizing shift in the inactivation parameter $V_{h1/2}$ and by a partial reduction in $W_{max}$; –20 and –25% for simulation of 5 µM and 10 µM RZ application, as described and justified previously (***Phillips and Rubin, 2019***). (**C**) Relationship between laser power and inspiratory burst frequency (top) and amplitude (bottom) of the integrated XII output with bath application of the pharmacological blockers of $I_{NaP}$ (TTX or RZ) or $I_{CAN}$ (9-phenanthrol); data are the same as shown in ***Figures 4C and 7D***. Data points and error bars are mean ± SEM.

The online version of this article includes the following source data for figure 8:

**Source data 1.** Related to ***Figure 8A–C***.

power curve is larger with riluzole than TTX application (as also shown in ***Phillips and Rubin, 2019***) as seen in the experimental data.

$I_{CAN}$ block ($g_{CAN}$ reduction) more strongly decreases population activity amplitude in the model simulations than does $I_{NaP}$ block and, in contrast to $I_{NaP}$ block, has little effect on population bursting frequency under baseline conditions but strongly augments bursting frequency at higher levels of excitability. As such, $I_{CAN}$ block results in an upward shift and steeper slope in the frequency tuning curve that extends the frequency range at the higher levels of population depolarization, consistent with the experimental results (***Figure 7***). Thus, the model predictions are qualitatively consistent with the experimental data for major features of excitatory preBötC network behavior.

# Materials and methods
## Model description and methods
As in the previous paper (***Phillips et al., 2019***), the preBötC model network is constructed with N=100 synaptically coupled excitatory neurons. Neurons are simulated with a single compartment described in the Hodgkin-Huxley formalism. The addition of $I_{ChR2}$ is a new feature of the current model compared to ***Phillips et al., 2019***. In the updated model, the membrane potential $V_m$ for each neuron is given by the following current balance equation:

$$C_m \frac{dV_m}{dt} + I_{Na} + I_K + I_{Leak} + I_{NaP} + I_{CAN} + I_{Ca} + I_{Syn} + I_{ChR2} = 0,$$

where $C_m$ is the membrane capacitance, $I_{Na}$, $I_K$, $I_{Leak}$, $I_{NaP}$, $I_{CAN}$, $I_{Ca}$, $I_{Syn}$, and $I_{ChR2}$ are ionic currents through sodium, potassium, leak, persistent sodium, calcium-activated non-selective cation, voltage-gated calcium, synaptic and ChR2 channels, respectively. For a full description of all currents other than $I_{ChR2}$, see ***Phillips et al., 2019***. The description of $I_{ChR2}$ was adapted from ***Williams et al., 2013*** and given by the following equation:

$$I_{ChR2} = g_{ChR2} \cdot G(V_m) \cdot (O_1 - \delta \cdot O_2) \cdot (V_m - E_{ChR2}),$$

where $g_{ChR2}$ is the maximal conductance, $O_1$ and $O_2$ are light-sensitive gating variables, $\delta$ is the ratio of the two open-state conductances, $V_m$ is the membrane potential, and $E_{ChR2}$ is the reversal potential for ChR2. All parameters are given in ***Table 1***. $G(V_m)$ is an empirically derived voltage-dependent function given by:

$$G(V_m) = \left[10.6408 - 14.6408 \cdot e^{-V_m/42.7671}\right] / V_m.$$

ChR2 is simulated as a four-state Markov channel with two open $O_1$ and $O_2$ and two closed states $C_1$ and $C_2$ (***Figure 9***). Transitions from $C_1$ to

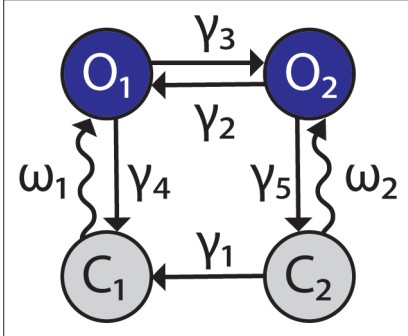

**Figure 9.** Channelrhodopsin-2 (ChR2) channel configuration. The ChR2 channel activation dynamics are described with a four-state Markov model, based on ***Williams et al., 2013***. Transition rates between states are represented by variables $\gamma_{1-5}$ and the light-dependent variables $\omega_{1,2}$. ***Figure 9*** has been adapted from Figure 1B in ***Williams et al., 2013***.

**Table 1.** Updated model parameters.

| Channel | Parameters |
|---|---|
| $I_{Na}$ | $g_{Na} = 150.0\ nS,\ E_{Na} = 55.0\ mV,$ <br> $V_{m_{1/2}} = 43.8\ mV,\ k_m = 6.0\ mV,$ <br> $V_{\tau m_{1/2}} = -43.8\ mV,\ k_{\tau_m} = 14.0\ mV,\ \tau_{m_{max}} = 0.25\ ms,$ <br> $V_{h_{1/2}} = 67.5\ mV,\ k_h = -10.8\ mV,$ <br> $V_{\tau h_{1/2}} = -67.5\ mV,\ k_{\tau_h} = 12.8\ mV,\ \tau_{h_{max}} = 8.46\ ms$ |
| $I_K$ | $g_K = 160.0\ nS,\ E_K = -94.0\ mV,$ <br> $A_\alpha = 0.01,\ B_\alpha = 44.0\ mV,\ \kappa_\alpha = 5.0\ mV,$ <br> $A_\beta = 0.17,\ B_\beta = 49.0\ mV,\ \kappa_\beta = 40.0\ mV$ |
| $I_{Leak}$ | $g_{Leak} = 2.5\ nS,\ E_{Leak} = -68.0\ mV$ |
| $I_{NaP}$ | $g_{NaP} \in [0.0, 5.0]\ nS,$ <br> $V_{m_{1/2}} = 47.1\ mV,\ k_m = 3.1\ mV,$ <br> $V_{\tau m_{1/2}} = -47.1\ mV,\ k_{\tau_m} = 6.2\ mV,\ \tau_{m_{max}} = 1.0\ ms,$ <br> $V_{h_{1/2}} = 60.0\ mV,\ k_h = -9.0\ mV,$ <br> $V_{\tau h_{1/2}} = -60.0\ mV,\ k_{\tau_h} = 9.0\ mV,\ \tau_{h_{max}} = 5000\ ms$ |
| $I_{CAN}$ | $g_{CAN} \in [0.5, 1.5]\ nS,\ E_{CAN} = 0.0\ mV,$ <br> $Ca_{1/2} = 0.00074\ mM,\ n = 0.97$ |
| $I_{Ca}$ | $\color{red}{g_{Ca} = 0.00175\ nS,}\ E_{Ca} = R \cdot T/F \cdot ln\left([Ca]_{out} / [Ca]_{in}\right),$ <br> $R = 8.314\ J/\left(mol \cdot K\right),\ T = 308.0\ K,$ <br> $F = 96.485\ kC/mol,\ [Ca]_{out} = 4.0\ mM,$ <br> $V_{m_{1/2}} = 27.5\ mV,\ k_m = 5.7\ mV,\ \tau_m = 0.5\ ms,$ <br> $V_{h_{1/2}} = 52.4\ mV,\ k_h = -5.2\ mV,\ \tau_h = 18.0\ ms$ |
| $Ca_{in}$ | $\alpha_{Ca} = 2.5 \cdot 10^{-5}\ mM/fC,\ \color{red}{P_{Ca} = 0.0225,}\ Ca_{min} = 1.0 \cdot 10^{-10}\ mM,\ \tau_{Ca} = 50.0\ ms$ |
| $I_{Syn}$ | $\color{red}{g_{Tonic} = 0.304\ nS,}\ E_{Syn} = -10.0\ mV,\ \tau_{Syn} = 5.0\ ms,$ <br> $P_{Syn} = 1,\ W_{max} = 0.01412\ nS$ |
| $I_{ChR2}$ | $g_{ChR2} = 1.19\ nS,\ E_{ChR2} = 0.0\ mV,\ \delta = 0.1,\ Irr_{ChR2} = [0, 2]\ mW,$ <br> $\sigma_{ret} = 1.0 \cdot 10^{-20}\ mm^2,\ \lambda = 470\ nm,\ \omega_{loss} = 0.77,$ <br> $h = 6.62607 \cdot 10^{-34}\ m^2 \cdot kg/s,$ <br> $C = 3.0 \cdot 10^8\ m/s,\ \tau_{ChR2} = 1.3\ ms$ |

$O_1$ and from $C_2$ to $O_2$ are light-sensitive and controlled by the parameter $Irr$ which represents optical power with units of milliwatts (mW).

The rates of change between states are given by the following differential equations:

$$\frac{dC_1}{dt} = \gamma_1 \cdot C_2 + \gamma_4 \cdot O_1 - \omega_1 \cdot C_1,$$

$$\frac{dC_2}{dt} = \gamma_5 \cdot O_2 - (\omega_2 + \gamma_1) \cdot C_2,$$

$$\frac{dO_1}{dt} = \omega_1 \cdot C_1 - (\gamma_4 + \gamma_3) \cdot O_1 + \gamma_2 \cdot O_2,$$

$$\frac{dO_2}{dt} = \omega_2 \cdot C_2 - (\gamma_5 + \gamma_2) \cdot O_2 + \gamma_3 \cdot O_1,$$

which are consistent with the condition that

$$O_1 + O_2 + C_1 + C_2 = 1.$$

The empirically estimated transition rate (ms$^{-1}$) parameters $\omega_1$, $\omega_2$, $\gamma_1$, $\gamma_2$, $\gamma_3$, $\gamma_4$, and $\gamma_5$ are defined as follows:

$$\omega_1 = 0.8535 \cdot F \cdot P,$$

$$\omega_2 = 0.14 \cdot F \cdot P,$$

$$\gamma_1 = 4.34 \cdot 10^5 \cdot e^{-0.0211539274 \cdot V_m},$$

$$\gamma_2 = 0.008 + 0.004 \cdot ln\left(1 + \frac{Irr}{0.024}\right),$$

$$\gamma_3 = 0.011 + 0.005 \cdot ln\left(1 + \frac{Irr}{0.024}\right),$$

$$\gamma_4 = 0.075 + 0.043 \cdot \tanh\left(\frac{V_m + 20}{-20}\right),$$

and

$$\gamma_5 = 0.05.$$

$F$ and $P$ represent the photon flux (number of photons per molecule per second) and time- and irradiance-dependent activation function for ChR2, respectively. These quantities take the form

$$F = \sigma_{ret} \cdot Irr \cdot \lambda / \left(\omega_{loss} \cdot h \cdot C\right),$$

$$\frac{dp}{dt} = \left[0.5 \cdot \left(1 + tanh\left(120 \cdot \left(Irr - 0.1\right)\right)\right) - p\right] / \tau_{ChR2},$$

where $\sigma_{ret}$ is the absorption cross-section, $\lambda$ is the wavelength of max absorption, $\omega_{loss}$ is a scaling factor for loss of photons due to scattering/absorption, $h$ is the Planks constant, $C$ is the speed of light, and $\tau_{ChR2}$ is the time constant of ChR2 activation. Model parameter values are given in *Table 1*.

## Model tuning

Initial model predictions presented in *Figure 1* were generated without modification of the initial model tuning presented in *Phillips et al., 2019*. To match experimental data (*Figure 8*), parameters were slightly modified from the original model. The updated list of parameters is given in *Table 1*, where parameter adjustments are indicated with red font.

## Integration methods

All simulations were performed locally on an eight-core computer running the Ubuntu 20.04 operating system. Simulation software was custom written in C++ and compiled with g++ version 9.3.0 (source code is provided as *Source code 1*). Numerical integration was performed using the exponential Euler method with a fixed step-size (dt) of 0.025 ms. In all simulations, the first 50 s of simulation time was discarded to allow for the decay of any initial condition-dependent transients.

## Experimental methods

### Animal procedures

All animal procedures were approved by the Animal Care and Use Committee of the National Institute of Neurological Disorders and Stroke (Animal Study Proposal number: 1154–21).

### Cre-dependent triple transgenic mouse model for optogenetic manipulation of VgluT2-expressing glutamatergic neurons

We have previously established and histologically validated the Cre-dependent double transgenic mouse line (VgluT2-tdTomato) to study population-specific roles of VgluT2-expressing and red fluorescent protein-labeled glutamatergic preBötC neurons (*Koizumi et al., 2016*). This VgluT2-tdTomato mouse line was produced using a *Slc17a6^(tm2(cre)Lowl)*/J knock-in Cre-driver mice strain (VgluT2-ires-Cre; IMSR Cat No. JAX: 016963, the Jackson Laboratory, Bar Harbor, ME), in which Cre-recombinase activity is under control by the VgluT2 promoter. This line was crossed with a Cre-dependent tdTomato reporter mouse strain (*B6;Cg-Gt[ROSA]26Sor^(tm9(CAG-tdTomato)Hze)*, Rosa-CAG-LSL-tdTomato-WPRE, IMSR Cat No. JAX: 007909, the Jackson Laboratory) to obtain offspring expressing the red fluorescent protein variant tdTomato in Cre-expressing VgluT2-positive (glutamatergic) neurons. The Cre-dependent triple transgenic mouse line VgluT2-tdTomato-ChR2-EFYP, which expresses ChR2-EYFP fusion protein and tdTomato reporter fluorescent protein in VgluT2-positive neurons was used for optogenetic photostimulation experiments and in some experiments for imaging glutamatergic neurons for targeted whole-cell patch-clamp electrophysiological recording from these neurons. This transgenic line was obtained by crossing the validated VgluT2-tdTomato line with the Cre-dependent optogenetic mouse strain (*B6;129S-Gt[ROSA]26Sor^(tm32(CAG-COP4*H134R/EYFP)Hze)*/J, the Jackson Laboratory).

## Rhythmically active medullary slice preparations in vitro

We performed optogenetic experiments combined with whole-cell patch-clamp or neural population electrophysiological recordings and pharmacological manipulations in rhythmically active in vitro medullary slice preparations (300–400 µm thick) from neonatal (postnatal day 3 [P3] to P8) VgluT2-tdTomato-ChR2 mice of either sex. The slice was superfused (4 ml/min) with artificial cerebrospinal fluid (ACSF) in a recording chamber (0.2 ml) mounted on the stage of an upright laser scanning microscope (TCS SP5 II MP, Leica, Buffalo Grove, IL). The ACSF contained the following (in mM): 124 NaCl, 25 NaHCO$_3$, 3 KCl, 1.5 CaCl$_2$, 1.0 MgSO$_4$, 0.5 NaH$_2$PO$_4$, 30 D-glucose equilibrated with 95% O$_2$ and 5% CO$_2$ (pH=7.35–7.40 at 27°C). During experiments, rhythmic respiratory network activity was maintained by elevating the superfusate K$^+$ concentration to 8–9 mM.

## Pharmacological reagents

TTX (Cat No. 1078, Tocris) at low concentration (5–20 nM), riluzole (5–20 µM; Cat No. R116, Sigma), and 9-phenanthrol (50 uM)—a specific inhibitor of TRPM4 channels at this concentration and putative blocker of I$_{CAN}$ (Cat No. 648492, EMD Millipore Corp., Billerica, MA), were applied to the bathing solution of the in vitro slice preparations.

## Electrophysiological recordings of preBötC and XII inspiratory activity

We recorded motor outputs in vitro from XII nerve rootlets with fire-polished glass suction electrodes (50–100 µm inner diameter) to monitor inspiratory XII motoneuron population activity. In addition, to monitor preBötC inspiratory population activity directly, we performed macro-patch recordings by applying a fire-polished glass pipette (150–300 µm inner diameter) directly on the surface of the preBötC region. Electrophysiological signals in all cases were amplified (50,000–100,000 X; CyberAmp 380, Molecular Devices, Union City, CA), band-pass filtered (0.3–2 kHz), digitized (10 kHz) with an AD converter (PowerLab, AD Instruments, Inc, Colorado Springs, CO or Cambridge Electronics Design, Cambridge, UK), and then rectified and integrated digitally with Chart software (AD Instruments) or Spike2 software (Cambridge Electronics Design).

Whole-cell voltage- and current-clamp data were recorded with a HEKA EPC-9 patch-clamp amplifier (HEKA Electronics Inc, Mahone Bay, Nova Scotia, Canada) controlled by PatchMaster software (HEKA; 2.9 kHz low-pass filter, sampled at 10 kHz). Whole-cell recording electrodes (borosilicate glass pipette, 4–6 MΩ), positioned with a three-dimensional micromanipulator (Scientifica, East Sussex, UK), contained the following (in mM): 130.0 K-gluconate, 5.0 Na-gluconate, 3.0 NaCl, 10.0 HEPES (4-(2-hydroxyethyl)-1-piperazineethanesulphonic acid) buffer, 4.0-mg ATP, 0.3 Na GTP, and 4.0 sodium phosphocreatine, pH 7.3 adjusted with KOH. In all cases, measured potentials were corrected for the liquid junction potential (–10 mV). Series resistance was compensated on-line by ~80%, and the compensation was periodically readjusted.

We identified, under current-clamp recording, optically imaged (below) preBötC VgluT2-tdTomato expressing inspiratory neurons that have intrinsic voltage-dependent oscillatory bursting properties (endogenous burster neurons; *Koizumi et al., 2013*). As shown previously (*Koizumi et al., 2013*), these endogenous burster neurons exhibited 'ectopic' bursts that are not synchronized with inspiratory XII activity in addition to exhibiting inspiratory bursts synchronized with this inspiratory activity. Membrane depolarization of these neurons by applying steady current caused an increase of ectopic bursting frequency in a voltage-dependent manner as shown previously (*Koizumi et al., 2013*). In some experiments, we confirmed that these neurons exhibited intrinsic voltage-dependent rhythmic bursting behavior after blocking excitatory synaptic transmission by bath application of the non-NMDA glutamate receptor blocker, 6-cyano-7-nitroquinoxaline-2,3-dione disodium (CNQX) (20 µM; Sigma; also see *Koizumi et al., 2013*).

Whole-cell voltage-clamp recording from optically identified VgluT2-tdTomato expressing inspiratory neurons was used to obtain neuronal I-V relationships by applying slow voltage ramps (30 mV/s; –100 to +10 mV) and we computed TTX- and riluzole-sensitive I$_{NaP}$ by subtracting I-V curves obtained before and after block of I$_{NaP}$ with TTX or riluzole. Data acquisition and analyses were performed with PatchMaster and Igor Pro (Wavemetrics) software (*Koizumi and Smith, 2008*). Values of inspiratory neuronal membrane capacitance (C$_m$) and maximum I$_{NaP}$ conductance (g$_{NaP}$), were measured from voltage-clamp data as previously described (*Koizumi and Smith, 2008*).

## Optical imaging of preBötC glutamatergic neurons

Optical two-photon live imaging of tdTomato-ChR2-EYFP-co-labeled neurons in the slice preparations was performed to verify fluorescent protein expression and target VgluT2-positive preBötC neurons for whole-cell recording. Images were obtained with a Leica multi-photon laser scanning upright microscope (TCS SP5 II MP with DM6000 CFS system, LAS AF software, 20× water-immersion objective, N.A. 1.0, Leica; and 560 nm beam splitter, emission filter 525/50, Semrock). A two-photon Ti:Sapphire pulsed laser (MaiTai, Spectra Physics, Mountain View, CA) was used at 800–880 nm with DeepSee predispersion compensation. The laser for two-photon fluorescence imaging was simultaneously used for transmission bright-field illumination to obtain a Dodt gradient contrast structural imaging to provide fluorescence and structural images matched to pixels. These images allowed us to accurately place a patch pipette on the tdTomato-labeled VgluT2-positive neurons to functionally identify preBötC inspiratory glutamatergic neurons that were active in phase with XII inspiratory activity.

## Photostimulation of preBötC glutamatergic neuron population

Laser illumination for optogenetic experiments was performed with a blue laser (473 nm; OptoDuet Laser, IkeCool, Los Angeles, CA), and laser power (0.25–5.0 mW) was measured with an optical power and energy meter (PM100D, ThorLabs, Newton, NJ). Illumination epochs (sustained 20 Hz, 20 ms pulses, variable duration pulse trains) were controlled by a pulse stimulator (Master-8, A.M.P.I., Jerusalem, Israel). Optical fibers, from a bifurcated fiberoptic patch cable, each terminated by an optical cannula (100 µm diameter, ThorLabs), were positioned bilaterally on the surface of the preBötC in the in vitro rhythmically active slice preparations.

## Signal analysis of respiratory parameters and statistics

All digitized electrophysiological signals were analyzed off-line to extract respiratory parameters (inspiratory burst frequency and amplitude) from the smoothed integrated XII nerve or preBötC neuronal population inspiratory activities (200ms window moving average) performed with Chart software (AD Instruments), Spike2 software (Cambridge Electronics Design), and Igor Pro (WaveMetrics) software. Following automated peak detection, burst period was computed to obtain the inspiratory burst frequency. Inspiratory burst amplitude was calculated as the difference between the signal peak value and the local baseline value. To obtain steady-state perturbations, we analyzed the respiratory parameters during laser illumination excluding the first and last 5–10 s of laser illumination. Power spectrum analyses were performed using MATLAB software ver. R2020b (MathWorks, Natick, MA, USA). Power spectra were calculated as a squared magnitude of the Fast Fourier Transform of the rectified preBötC activity for 30 s segments of the recordings during baseline activity, photostimulation, and recovery periods. The oscillatory component in preBötC activity was quantified as a ratio of the band power of the signal in the 0–3 Hz frequency range to the band power of the signal in the 3–6 Hz frequency range. For statistical analyses, respiratory parameters during laser illumination were normalized to the control value calculated as an average from ~30 inspiratory bursts before laser illumination. The normality of data was assessed both visually (quantile vs quantile plots) and through the Shapiro-Wilk normality test. Statistical significance ($p < 0.05$) was determined with non-parametric Wilcoxon matched-pairs signed rank test or Kolmogorov-Smirnov test when comparing two groups, and two-way ANOVA test for comparing multiple groups in conjunction with post hoc Tukey's HSD test for pairwise comparisons (Prism, GraphPad software LLC), and summary data are presented as the mean ± SEM.

# Discussion

In these new combined experimental and modeling studies, we have advanced our understanding of neuronal and circuit biophysical mechanisms generating the rhythm and amplitude of inspiratory activity in the brainstem preBötC inspiratory oscillator in vitro. These studies were designed to further experimentally test predictions of our recent computational model (*Phillips et al., 2019*) of preBötC excitatory circuits that incorporate rhythm- and amplitude-generating biophysical mechanisms, relying, respectively, on a neuronal subthreshold-activating, slowly inactivating $I_{NaP}$, and an $I_{CAN}$, mediated by TRPM4 channels coupled to intracellular calcium dynamics. Our model explains how the

functions of generating the rhythm and amplitude of inspiratory oscillations in preBötC excitatory circuits involve distinct biophysical mechanisms. In essence, the model advances the concepts that (1) a subset of excitatory circuit neurons whose rhythmic bursting in vitro is critically dependent on $I_{NaP}$ forms an excitatory neuronal kernel for inspiratory rhythm generation, and (2) excitatory synaptic drive from the rhythmogenic kernel population is critically amplified by $I_{CAN}$ activation in recruited and interconnected preBötC excitatory neurons in the network to generate the amplitude of population activity.

The model predicts that $I_{NaP}$ is critical for rhythm generation and confers voltage-dependent rhythmic burst frequency control in vitro, which provides a mechanism for frequency tuning over a wide dynamic range defined by the frequency tuning curve for the rhythmogenic population (i.e. the relationship between applied current/baseline membrane potential across the network and network bursting frequency). Accordingly, reducing neuronal $g_{NaP}$ decreases the population-level bursting frequency with a weaker reduction in amplitude, reduces the frequency dynamic range, and, at sufficiently low levels of $g_{NaP}$, terminates rhythm generation under baseline and other conditions of network excitability in vitro, as also shown in *Phillips and Rubin, 2019*. In contrast, $I_{CAN}$ in the model is essential for generating the amplitude of rhythmic output but not for rhythm generation. As a result, reducing $g_{CAN}$ strongly decreases population activity amplitude and has little effect on population bursting frequency under baseline conditions in vitro, but strongly augments bursting frequency, due to a shift in the frequency tuning curve that extends the frequency range, at higher levels of population excitation. These predicted opposing effects of $I_{NaP}$ and $I_{CAN}$ attenuation on the relationship between network excitability and preBötC population rhythmic activity provided a clear basis for model testing, and our experimental results showed an overall strong agreement with the model predictions.

To experimentally test these predictions in the rhythmically active in vitro preBötC network in slices from transgenic mice, we used a combination of electrophysiological analyses, pharmacological perturbations of $I_{NaP}$ or $I_{CAN}$, and selective optogenetic manipulations of the preBötC excitatory (glutamatergic) population involved. Our application of optogenetic photostimulation of preBötC glutamatergic neurons to examine the population activity frequency tuning curves under these different conditions of $I_{NaP}$ or $I_{CAN}$ attenuation provided an effective way to probe for underlying mechanisms. We performed new simulations to mimic the optogenetic manipulations under conditions of pharmacological perturbations of $I_{NaP}$ or $I_{CAN}$. $I_{NaP}$ and $I_{CAN}$ block in the experimental data yield shifts in the preBötC network burst frequency/amplitude vs laser power curves (*Figure 8*) in the same direction, and of the same relative magnitudes, as predicted by our simulations (*Figures 1 and 8*). This correspondence strongly supports the main hypothesis from the model that $I_{NaP}$ and $I_{CAN}$ mechanistically underlie preBötC rhythm and inspiratory burst pattern generation, respectively. The similarities between the model predictions and experimental results support the basic concepts about inspiratory rhythm and burst pattern generation from the model.

## Extensions and limitations of the model

The assumptions and neurobiological simplifications of the original model were previously elaborated in *Phillips et al., 2019*. We have extended the model to allow comparisons of our new experimental data and model simulation results. The updates included incorporating a $I_{ChR2}$ to represent $I_{App}$ for the entire excitatory neuron population and to mimic effects of photo-induced neuronal depolarization. We modeled ChR2 by a four-state Markov channel based on channel biophysical representations in the literature (*Williams et al., 2013*). We found that this biophysical model could be parameterized to closely match our experimental data on relations between laser power and membrane depolarization of identified preBötC inspiratory glutamatergic neurons. Furthermore, we found that this channel model parameterized from our cellular-level data yielded frequency tuning curves for the bilateral excitatory population under baseline conditions and with pharmacological perturbations that were in close agreement with the experimentally measured tuning curves. We note that in the model, the assumption is that all excitatory network neurons have the same level of depolarization at a given laser power. In the experimental in vitro slice preparations, this may not occur if the neuronal expression levels of ChR2 are non-uniform. We have used transgenic strategies in mice to drive neuronal expression of ChR2 throughout the population of glutamatergic neurons, which has the advantage that there should be efficiency in coverage of this population in terms of channel expression. In support of this assumption, our single neuron electrophysiological data show a small SEM in the relations between

neuronal depolarization and laser power, but it is not possible to fully quantify the uniformity of ChR2 expression levels across the entire population of glutamatergic inspiratory neurons.

We also note that the transgenic strategy used here may result in expression of ChR2 in glutamatergic terminals, including on preBötC inhibitory neurons that are proposed to interact with the excitatory neurons (e.g. *Ausborn et al., 2018*; *Ashhad and Feldman, 2020*), as well as fibers of passage from non-preBötC neurons, which could result in off-target effects. However, as discussed above, we have calibrated the ChR2 model to match the levels of depolarization at a given laser power that we have measured experimentally at the neuronal level. These measurements represent the total amount of neuronal depolarization which would include any contributions from photostimulated glutamatergic fibers and terminals. Thus, our ChR2 model and simulated photostimulation experiments adequately account for all sources contributing to neuronal depolarization and such off-target effects related to ChR2 expression, if present, would not impact the conclusions of this study.

To compare the model behavior with data from the pharmacological manipulations, we simulated blockade of $I_{NaP}$ by TTX or $I_{CAN}$ inhibition by 9-phenanthrol by reducing the channel conductances $g_{NaP}$ and $g_{CAN}$, respectively. Blockade of $I_{NaP}$ by riluzole was simulated by a hyperpolarizing shift in the inactivation parameter $V_{h1/2}$ and by a partial reduction in $W_{max}$ (–20% and –25% for simulation of 5 μM and 10 μM riluzole), as described and justified previously (*Phillips and Rubin, 2019*) to take into account known or proposed pharmacological actions of riluzole from the literature. With this approach, the simulation results were directionally consistent with the data showing that $I_{NaP}$ attenuation, by either low concentrations of TTX or by riluzole, and $I_{CAN}$ blockade have opposite effects on the relationship between network excitability controlled by photostimulation and preBötC population burst frequency. We verified the presence of $I_{NaP}$ in preBötC glutamatergic inspiratory neurons from voltage-clamp measurements and demonstrated partial and complete block of this current by TTX or riluzole under our in vitro experimental conditions. A difference between the modeling and experimental results that may represent a limitation of the model is that the model network is more sensitive to $I_{NaP}$ block in terms of rhythm perturbation than suggested by the experimental results. In both model simulations and experiments, partial or complete block of this current can terminate rhythm generation; however, rhythm generation can stop at a smaller percentage reduction in $I_{NaP}$ in the model, where it is assumed that $I_{NaP}$ in all network neurons is uniformly reduced by the same amount at a given simulated percent block. This uniformity may not be the case in the in vitro network, where heterogeneity of cellular biophysical properties and spatial variability of pharmacological actions associated with drug penetration problems in the slice could cause non-uniformity of block at a given drug concentration.

Furthermore, the biophysical and pharmacological properties of $I_{NaP}$ in preBötC inspiratory neurons are not fully characterized. Our recent biophysical characterization of $I_{NaP}$ for these neurons has confirmed that this current is slowly inactivating (*Yamanishi et al., 2018*), which is a basic assumption of the present and previous models (*Phillips et al., 2019*), although the details of the voltage-dependent kinetic properties are more complex than represented by the first-order kinetics in the models. Moreover, some additional currents or dynamical processes (e.g. inhibitory currents from local circuit connections, synaptic depression, burst terminating currents, ionic dynamics; *Thoby-Brisson et al., 2000*; *Hayes et al., 2008*; *Jasinski et al., 2013*; *Kottick and Del Negro, 2015*; *Phillips et al., 2018*; *Baertsch et al., 2018*; *Juárez-Vidales et al., 2021*; *Abdulla et al., 2022*; *Revill et al., 2021*), which are not considered in the model, may play a role in augmenting/shaping inspiratory bursting, contribute to robustness of the rhythm such as the electrogenic transmembrane $Na^+/K^+$-ATPase pump (*Jasinski et al., 2013*), as well as affect preBötC dynamics on longer time scales than those investigated here. For example, the current model does not predict the minutes long decrease in network burst frequency that may occur following optogenetic stimulation in some experiments (see *Figure 3*, but also see *Figure 7B*, where this effect is less strong). Similar activity-dependent inhibitory effects in mammalian motor networks and tadpole locomotion neurocircuits have been shown to be mediated by the $Na^+/K^+$-ATPase pump (*Picton et al., 2017*; *Picton et al., 2018*). Therefore, capturing this feature in our model may require inclusion of the $Na^+/K^+$-ATPase pump which is beyond the scope of this study and, along with the exploration of other possible mechanisms that may impact or shape preBötC inspiratory bursting, is left for future investigations. Nonetheless, with the kinetic and pharmacological properties incorporated in our model, the directional trends in the experimental results for the various pharmacological manipulations of $I_{NaP}$ and $I_{CAN}$ are correctly predicted by the model simulations

with very minor additional adjustments of key parameters relative to the initial model developed in *Phillips et al., 2019*.

## Further insights into mechanisms of rhythm and burst amplitude generation in preBötC excitatory circuits in vitro

Many previous experimental and theoretical analyses have focused on the rhythmogenic mechanisms operating under in vitro conditions in rhythmically active slices to provide insight into the biophysical and circuit processes involved, which as discussed in a number of reviews, has potential relevance for rhythm generation in vivo (*Feldman and Del Negro, 2006*; *Richter and Smith, 2014*; *Phillips et al., 2019*). There is an agreement that preBötC circuits have intrinsic autorhythmic properties, particularly because when isolated in slices in vitro, these circuits continue under appropriate conditions of excitability to generate rhythmic activity that drives behaviorally relevant inspiratory hypoglossal motoneuronal output. However, there is currently no consensus about the underlying rhythmogenic mechanisms. The endogenous rhythmic activity in vitro has been suggested to arise from various cellular and circuit biophysical mechanisms, including from a subset of intrinsically bursting neurons which, through excitatory synaptic interactions, recruit and synchronize neurons within the network (pacemaker-network models; *Rekling and Feldman, 1998*; *Ramirez et al., 2004*; *Toporikova and Butera, 2011*), or as an emergent network property involving recurrent excitation (*Jasinski et al., 2013*) and/or synaptic depression (group pacemaker model; *Rubin et al., 2009a*). More recent 'burstlet' models for rhythm generation emphasize how rhythm emerges in low excitability states due to synchronization of subsets of excitatory bursting neurons (*Kam et al., 2013*; *Del Negro et al., 2018*). Extensions of our current model (*Phillips and Rubin, 2022*) with our proposed $I_{NaP}$ and $I_{CAN}$ biophysical mechanisms for rhythm and activity amplitude generation can explain the generation of burstlets and synchronized population activity.

The present experimental results are consistent with previous findings indicating that the subthreshold-activating, slowly inactivating $I_{NaP}$ is a critical neuronal conductance for inspiratory rhythm generation and neuronal bursting in vitro (*Koizumi and Smith, 2008*; *Yamanishi et al., 2018*). As our previous experimental and modeling results have indicated, the voltage-dependent and kinetic properties of these channels are fully capable of orchestrating rhythmic bursting at neuronal (e.g. *Yamanishi et al., 2018*) and excitatory neuron population levels (*Koizumi et al., 2016*). In the experiments blocking $I_{NaP}$, we tested by power spectral analyses for rhythmogenic mechanisms that do not depend on $I_{NaP}$ after fully blocking this conductance as verified by our cellular-level measurements. These experiments were designed to determine if there are additional rhythmogenic mechanisms inherent within the excitatory network that can emerge during graded levels of tonic network excitation as controlled in our experiments by sustained bilateral photostimulation. We did not find any coherent rhythmic population activity under these conditions that would indicate the presence of other important rhythmogenic mechanisms capable of producing rhythmic population bursting on any time scale, which is consistent with our model. These results do not support the concept that rhythm generation under in vitro conditions is an emergent network property through recurrent excitation (*Jasinski et al., 2013*) and/or synaptic depression (group pacemaker model; *Rubin et al., 2009a*). Such emergent rhythms are theoretically possible as shown by previous modeling studies (*Rubin et al., 2009a*; *Jasinski et al., 2013*; *Guerrier et al., 2015*), but the regenerative population burst-generating and burst-terminating mechanisms incorporated in these models are apparently not sufficiently expressed in the in vitro preBötC network.

We note that our new measurements indicate that while $I_{CAN}$/TRPM4 activation is not involved in rhythm generation under baseline conditions of network excitability, since inhibiting this current does not significantly affect the rhythm, we also show that reducing this current does shift the frequency tuning curve at higher levels of network excitation. This occurs because activation of this current augments excitatory synaptic interactions, which can affect network bursting frequency. As discussed above, these observed effects of reducing $I_{CAN}$/TRPM4 on bursting frequency at elevated levels of network excitation, which we were able to reveal with our photostimulation paradigm, is a basic feature and entirely consistent with predictions of our model.

Our new experimental results confirm our previous experimental and modeling results indicating that endogenous activation of $I_{CAN}$/TRPM4 is critically involved in generating the amplitude of population activity but not the rhythm, which is also consistent with results from a genetic knockdown study

of TRPM4 channels in preBötC neurons in vivo (*Picardo et al., 2019*). The correspondence between the experimental data and model predictions supports the concept in the model that activation of $I_{CAN}$ is largely due to synaptically activated sources of neuronal $Ca^{2+}$ flux such that $I_{CAN}$ contributes to the excitatory inspiratory drive potential and regulates inspiratory burst amplitude by augmenting the excitatory synaptic current. Our data also show that $I_{NaP}$ is involved, to a smaller degree, in generating the amplitude of rhythmic population activity at a given level of excitatory network excitation. This is due to the subthreshold activation of $I_{NaP}$ and its voltage-dependent amplification of synaptic drive; indeed, application of riluzole, which is thought to impact synaptic transmission, affected population amplitude much more than application of TTX (*Figures 4 and 8*). This contribution to amplitude is smaller than that from $I_{CAN}$ /TRPM4 activation, however. In general, our results support the concept from our original model that $I_{CAN}$ activation in a subpopulation of preBötC excitatory neurons is critically involved in amplifying synaptic drive from a subset of neurons whose rhythmic bursting is critically dependent on $I_{NaP}$ . This latter subpopulation forms the kernel for rhythm generation in vitro.

## Previous pharmacological studies and proposed roles of $I_{NaP}$ in preBötC inspiratory network rhythm generation

The role of $I_{NaP}$ in rhythm generation within preBötC circuits is highly debated in the field with diverse and contradictory experimental results (*Del Negro et al., 2002*; *Peña et al., 2004*; *Ramirez et al., 2004*; *Del Negro et al., 2005*; *Feldman and Del Negro, 2006*; *Smith et al., 2007*; *Pace et al., 2007*; *Koizumi and Smith, 2008*; *Ashhad and Feldman, 2020*). Pharmacological studies (*Smith et al., 2007*; *Koizumi and Smith, 2008*) using riluzole or low concentrations of TTX to block $I_{NaP}$ in preBötC neurons/circuits, as in the present studies, demonstrated a large reduction of preBötC inspiratory bursting frequency at cellular and network levels, with relatively smaller reductions of inspiratory network activity amplitude; fully blocking $I_{NaP}$ completely eliminated inspiratory rhythm generation within isolated preBötC circuits in vitro. These experimental results are consistent with the computational hypothesis presented here and earlier that $I_{NaP}$ is an essential biophysical component of inspiratory rhythm generation in vitro (*Butera et al., 1999a*; *Butera et al., 1999b*; *Smith et al., 2000*; *Phillips et al., 2019*; *Phillips and Rubin, 2019*).

However, this hypothesis has been challenged by some previous studies (*Del Negro et al., 2002*; *Peña et al., 2004*; *Pace et al., 2007*). In particular, *Pace et al., 2007* found that microinjecting 10 µM riluzole or 20 nM TTX directly into the raphe obscurus but not the preBötC stopped inspiratory rhythm generation in vitro. Moreover, preBötC rhythm generation could be restarted by bath application of 0.5 µM substance P following bath application and microinjection of riluzole. These results suggest that bath application of riluzole or TTX does not compromise the fundamental mechanism(s) of preBötC rhythmogenesis but rather the level of neuronal excitability via off-target effects in the raphe obscurus. These results have not been reproduced, however, and in a follow-up study, *Koizumi and Smith, 2008* observed that bath application or bilateral microinfusion in the preBötC of 10 µM riluzole or 20 nM TTX reliably abolished preBötC rhythm generation. The same is true for bath application of TTX in the in vitro preBötC island preparations that do not contain the raphe obscurus. Moreover, bath application of 1 µM substance P could only restart the preBötC rhythm when $I_{NaP}$ block was incomplete. The present study is also a direct test of the conclusions reached by *Pace et al., 2007*. If, as suggested by *Pace et al., 2007*, bath application of TTX or riluzole impacts the inspiratory rhythm by reducing preBötC excitability rather than by affecting the essential mechanism(s) of rhythm generation, then increasing preBötC excitability via optogenetic stimulation should restart the rhythm even after complete $I_{NaP}$ blockade. Our results show that, to the contrary, the preBötC is incapable of generating rhythmic output after complete $I_{NaP}$ block even under optogenetic stimulation (*Figures 4 and 5*), demonstrating that $I_{NaP}$ is essential for preBötC rhythm generation in this reduced in vitro preparation.

The off-target effects of riluzole are also a concern when interpreting its impact on preBötC rhythm and pattern generation. Specifically, at the concentrations used in these experiments ($\leq$20 µM) riluzole is reported to attenuate excitatory synaptic transmission (*MacIver et al., 1996*; *Bellingham, 2011*). Reducing the strength of excitatory synaptic transmission has previously been shown to decrease preBötC network burst amplitude and slightly increase frequency in computational models (*Phillips et al., 2019*; *Phillips and Rubin, 2019*) and experimental studies (*Johnson et al., 1994*). Moreover, *Phillips et al., 2019* showed that a 7–30% reduction in the strength of excitatory synaptic transmission was required to match the dose-dependent reduction of network burst amplitude seen with

progressive riluzole infusion into the preBötC in the in vitro slice preparations. In the current study, the larger downward shift in the network amplitude vs laser power curve seen in the model with simulated riluzole application (compared to TTX), as also seen in our in vitro slice preparations (*Figure 8B–C*), is due to the 20% (at 5 µM) to 25% (at 10 µM) attenuation of excitatory synaptic conductances. Therefore, the subtly different effects of riluzole and TTX application on network burst amplitude in the current study support the idea that riluzole diminishes excitatory synaptic transmission by approximately these factors.

If $I_{NaP}$ is critical for rhythm generation in the preBötC, then what can explain the findings (*Del Negro et al., 2002*; *Peña et al., 2004*; *Pace et al., 2007*) that seem to contradict this conclusion? Although more experimental work is needed to provide a definitive answer, possible explanations can be derived from computational modeling by considering non-uniformity of $I_{NaP}$ blockade, in vitro slice thickness, and the order in which neurons (patterning vs rhythmogenic neurons) in the network are affected (*Phillips and Rubin, 2019*). Penetration of bath applied or microinjected $I_{NaP}$ blockers in the in vitro slice preparations depends on passive diffusion. Therefore, the magnitude and progression of $I_{NaP}$ block across the slice will not be uniform and drug penetration may be incomplete in thicker slices, even with microinjection within the preBötC; limited drug diffusion could explain why microinjection of TTX or riluzole into the preBötC may fail to stop rhythm generation (*Pace et al., 2007*). Moreover, the preBötC network dynamically expands/contracts in the rostralcaudal axis (*Baertsch et al., 2019*), which suggests that rhythmogenic neurons are preferentially located near the center of a larger population of pattern forming inspiratory neurons. This observation leads to the prediction that $I_{NaP}$ block in thick slices via bath application of TTX or riluzole will primarily affect the amplitude of the inspiratory rhythm rather than its frequency and persistence, which matches the findings of in vitro experiments using thick slices (*Del Negro et al., 2002*; *Peña et al., 2004*; *Pace et al., 2007*). Another factor that impacts dynamics in thicker slices than those used in our studies is that they contain not only the preBötC, but also other network components that may introduce additional mechanisms. Indeed, with more intact brainstem respiratory pattern generation networks that exhibit more aspects of the full inspiratory and expiratory neuronal activity pattern in mature rodents, $I_{NaP}$ blockers do not disrupt rhythm generation (*Smith et al., 2007*; *Rubin et al., 2009b*; *Rubin and Smith, 2019*). In the more intact system, neuronal dynamics in the preBötC are controlled by more complex synaptic interactions, including inhibitory circuit interactions, with new rhythmogenic regimes (*Rubin et al., 2009b*; *Rubin and Smith, 2019*). Interestingly, modeling results suggest that there are network states and pharmacological conditions under which the inspiratory rhythm may be highly dependent upon $I_{NaP}$ in the more intact respiratory pattern generation network (*Phillips and Rubin, 2019*; *Rubin and Smith, 2019*), and these predictions require further experimental testing.

Regardless of this greater complexity, our present study has confirmed our previous results that $I_{NaP}$ is essential for inspiratory rhythm generation in the isolated preBötC excitatory circuits in vitro, and represents a major rhythmogenic mechanism in this reduced state of the respiratory network. An important prediction of our model (see *Phillips et al., 2019*, *Figure 9*) is that the subpopulation of excitatory neurons with $I_{NaP}$-dependent rhythmogenic properties forming the rhythmogenic kernel may be relatively small compared to the neuronal subpopulation(s) with the $I_{CAN}$-dependent mechanism that critically amplifies the rhythmic drive from the kernel to generate and control the amplitude of excitatory population activity. This prediction also requires direct experimental testing, such as by dynamic multicellular $Ca^{2+}$ imaging in glutamatergic neurons in vitro (*Koizumi et al., 2018*).

## Summary

Our exploitation of optogenetic control of glutamatergic neuron network excitability, in combination with specific pharmacological manipulations of neuronal conductances, has enabled rigorous testing of predictions of our previous model of preBötC excitatory circuits. The basic predictions of the model for cellular and network behavior under the experimental conditions tested show a strong overall agreement with our experimental results. This agreement advances our understanding of neuronal and circuit biophysical mechanisms generating the rhythm and amplitude of inspiratory activity in the brainstem preBötC inspiratory oscillator in vitro and demonstrates the predictive power of our model.

## Acknowledgements

This work was supported in part by the Intramural Research Program of the National Institutes of Health (NIH), National Institute of Neurological Disorders and Stroke (JS, HK), and grants from NSF DMS 1951095 (JR), NSF DMS 1724240 (JR, RSP), NIH R01 AT008632 (YM), NIH U01 EB021960 (YM) and Georgia State University B&B Seed Grant (YM).

## Additional information

### Competing interests
Jeffrey C Smith: Reviewing editor, eLife. The other authors declare that no competing interests exist.

### Funding

| Funder | Grant reference number | Author |
| --- | --- | --- |
| National Institute of Neurological Disorders and Stroke | Intramural Research Program | Hidehiko Koizumi Jeffrey C Smith |
| National Science Foundation | DMS 1951095 | Jonathan E Rubin |
| National Science Foundation | DMS 1724240 | Ryan S Phillips Jonathan E Rubin |
| National Institutes of Health | R01 AT008632 | Yaroslav I Molkov |
| National Institutes of Health | U01 EB021960 | Yaroslav I Molkov |
| Georgia State University | B&B Seed Grant | Yaroslav I Molkov |

The funders had no role in study design, data collection and interpretation, or the decision to submit the work for publication.

### Author contributions
Ryan S Phillips, Conceptualization, Data curation, Formal analysis, Writing – original draft, Writing – review and editing; Hidehiko Koizumi, Conceptualization, Data curation, Formal analysis, Writing – review and editing; Yaroslav I Molkov, Formal analysis, Funding acquisition, Writing – review and editing; Jonathan E Rubin, Funding acquisition, Supervision, Writing – review and editing; Jeffrey C Smith, Conceptualization, Funding acquisition, Supervision, Writing – original draft, Writing – review and editing

### Author ORCIDs
Ryan S Phillips ⓘ http://orcid.org/0000-0002-8570-2348
Hidehiko Koizumi ⓘ http://orcid.org/0000-0002-7747-3434
Yaroslav I Molkov ⓘ http://orcid.org/0000-0002-0862-1974
Jonathan E Rubin ⓘ http://orcid.org/0000-0002-1513-1551
Jeffrey C Smith ⓘ http://orcid.org/0000-0002-7676-4643

### Ethics
This study was performed in strict accordance with the recommendations in the Guide for the Care and Use of Laboratory Animals of the National Institutes of Health, the National Institute of Neurological Disorders and Stroke (Animal Study Proposal number: 1154-21).

### Decision letter and Author response
Decision letter https://doi.org/10.7554/eLife.74762.sa1
Author response https://doi.org/10.7554/eLife.74762.sa2

## Additional files

### Supplementary files

- Transparent reporting form
- Source code 1. Model source code.

### Data availability

All data generated or analyzed during this study are included in the manuscript and supporting source data files. All model parameters and equations are included in the manuscript and source code is provided.

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
