## [Editor Report]

This paper tests hypotheses for the role of INaP and ICAN in the preBötC, the region of the brainstem that generates inspiratory breathing rhythm, using optogenetic manipulation of local preBötC excitability and pharmacologic blockade of INaP and ICAN and tests resulting predictions about these currents using computational simulation. The paper will be of interest to respiratory researchers and all those interested in neuronal rhythm generation.

---

## [Decision Letter]

**Decision letter after peer review:**

Thank you for submitting your article "Predictions and experimental tests of a new biophysical model of the mammalian respiratory oscillator" for consideration by *eLife*. Your article has been reviewed by 3 peer reviewers, and the evaluation has been overseen by Ronald Calabrese as the Senior and Reviewing Editor. The following individuals involved in review of your submission have agreed to reveal their identity: Luis Rodrigo Hernandez-Miranda (Reviewer #1); Christopher Wilson (Reviewer #3).

Essential revisions:

1. The experimental data rest in large part on ChR2 providing essentially a step depolarization specifically and uniformly to preBotC glutamatergic neurons. However, ChR2 stimulation appears to produce some unexpected and unexplained effects of ChR2 stimulation that are not reproduced in the model and that may result from ChR2 expression in glutamatergic terminals from non-preBotC neurons and perhaps incomplete light activation of deeper neurons (off target effects). In particular there is concern that the post-stimulus inhibition appears to be a significant effect that is not replicated by the model and suggests that assumptions about ChR2 in the model require greater scrutiny. Absent further development of the model to encompass this phenomenon the claim that the experiments fully confirm model predictions should be scaled back.

2. The authors' attempts to resolve the conflicting data for the INaP necessity hypothesis in the Discussion overlooked experimental details in papers that counter the authors arguments. For example, Pace et al., (2007) showed that bilateral microinjection of riluzole or low concentrations of TTX into preBötC failed to stop the rhythm and that the pharmacological effects of these blockers could be explained by their effects on raphe excitability, which provides tonic excitatory drive to the preBötC. The authors propose that these conflicting results can be explained by differences in slice thickness and incomplete pharmacological penetration; however, the Pace paper specifically addressed this issue by microinjecting the drugs 100 μm below the surface. This could, of course, still lead to differences in penetration and there are still differences in the amount of network in the slice, but the discussion of thick vs. thin omits needed details here, and these effects can be addressed experimentally, in future. Absent such future experiments the conclusion that INaP is essential for must be removed and replaced by a more balanced conclusion.

3. Reviewer #3 asked for more detail about individual bursting neurons and their firing profiles because the relative expression of gNaP is important for endogenous bursting neurons. So, the authors would have a stronger argument if they included gNaP/Cm (as was done in the Koizumi et al., (2008)) and then showed what happens to those individual neurons when INaP and I_CAN are blocked. That would speak to questions re: variability in rhythm and give an idea of just how much INaP is present in the neurons recorded.

4. Please address the statistical issues brought up by the reviewers.

5. Please provide the missing details of the model requested by the reviewers.

6. Each reviewer provides detailed comments that will supplement and expand this summary.

*Reviewer #1 (Recommendations for the authors):*

In the current study, Phillips at al., experimentally tested three prediction that emerged from a previously published computational model (*eLife* 2019, 8:e41555): (1) the blockade of ICAN and INaP produces opposite effects on preBötC rhythmic activity; (2) ICAN is essential for preBötC rhythmogenesis; and (3) ICAN is key for generating the amplitude of respiratory rhythmic output. To do so, the authors used optogenetic/pharmacologic stimulation of the preBötC on mouse brainstem slices. These three predictions are, to a large extend, demonstrated with the new provided experimental data. Globally, the new findings reported by Phillips and colleagues foster our understanding on the elusive mechanisms that underline the generation of the respiratory (more precisely inspiratory) rhythm in mammals, which are of great interest for researchers working in respiratory physiology.

Comments

1. I have no problems with the model simulations/predictions nor with the findings of this carefully done work. Nevertheless, in my view, it is written in a highly technical manner that is not accessible to neuroscientists working on areas distinct to electrophysiology/computational modeling, which might preclude the full understanding of this very interesting study. Therefore, I would suggest to the authors to work a little bit on making this study more accessible to the large readership of *eLife*.

2. One aspect that could perhaps be discussed in this work is if Phillips's model could also consider the interconnections existing between left/right preBötC in rhythmogenesis and population activity amplitude. In other words, can the current computational model predict how ipsilateral changes in ICAN and INaP might alter population activity amplitude and rhythmogenesis on the contralateral preBötC?

*Reviewer #2 (Recommendations for the authors):*

Connectivity has been shown to be an important parameter in preBötC dynamics and was explored in the previous publication of this model, but the connectivity matrix/synaptic parameters are not described in this text and should be included.

Data from the model showing how optogenetic stimulation in the model compares to experimental results, particularly with respect to poststimulation membrane polarization and network effects, e.g., inhibition of rhythmicity following stimulation, should be presented.

In "Model Tuning", changed parameters are stated to be marked in red, but there is no red text; although, some values/terms do appear to be slightly bolded. Please use a clearer mark that is color-blind friendly to indicate updated parameters.

The rationale for using power spectrum analysis over analyzing the amplitude and frequency of preBötC activity is unclear. The physiological relevance of power in higher harmonic frequencies should be explained.

When discussing TRPM4 and ICAN, Picardo et al., 2019 PLOS Biology should be cited and discussed.

*Reviewer #3 (Recommendations for the authors):*

The manuscript is well-written and clear. The experiments are appropriate to test the hypotheses and the data is convincing. This manuscript is significant because it provides substantive evidence for the role of INaP in modulating breathing frequency and ICAN in altering amplitude with some interesting boundary conditions when ICAN and INaP are selectively blocked. Of particular value is the addition of a channelrhodopsin current (based on a Markov formalism) to the authors' previously published model.

I have no major concerns regarding the manuscript.

[Editors’ note: further revisions were suggested prior to acceptance, as described below.]

Thank you for resubmitting your work entitled "Predictions and experimental tests of a new biophysical model of the mammalian respiratory oscillator" for further consideration by *eLife*. Your revised article has been evaluated by Ronald Calabrese (Senior and Reviewing Editor).

The manuscript has been improved but there are some remaining issues that need to be addressed, as outlined below:

Please revise according to the reviewer Recommendations to the authors. Re-review will be by the reviewing editor.

*Reviewer #1 (Recommendations for the authors):*

The authors have adequately addressed all my comments and I appreciate the effort made to better describe this nice work to the broader readership of *eLife*. I congratulate the authors for their work.

*Reviewer #2 (Recommendations for the authors):*

In this revised manuscript, Phillips et al., respond to many of the reviewers’ concerns; however, there remain a few issues that should be addressed.

From Essential Revisions

1. Regarding limitations of the ChR2 experiments, the authors should mention and discuss the possibility that optogenetic stimulation of glutamatergic terminals may synaptically activate preBötC inhibitory neurons, thus altering excitation-inhibition balance (Ashhad and Feldman 2020). Could this mechanism also explain the post-stimulus inhibition?

2. At the end of the Discussion, in the Summary, and in the author’s response, the authors state that their results support the statement: “INaP is essential for rhythm generation in the reduced in vitro preparation used in this study.” A similar qualifier, like adding “in vitro”, should be placed wherever a statement about InaP being essential is made, especially in the abstract, second to last paragraph of Introduction, and first two paragraphs of the Discussion.

*Reviewer #3 (Recommendations for the authors):*

The authors have addressed the majority of my comments. My only remaining concern is their unwillingness to assess variability in their data since this seems a trivial “ask”---particularly if they choose perhaps the simplest metric of variability, coefficient of variation. Nonetheless, I still feel the manuscript is of value and advances the field and the authors have addressed the majority of concerns for each of the reviewers.

---

## [Author Response]

Essential revisions:1. The experimental data rest in large part on ChR2 providing essentially a step depolarization specifically and uniformly to preBotC glutamatergic neurons. However, ChR2 stimulation appears to produce some unexpected and unexplained effects of ChR2 stimulation that are not reproduced in the model and that may result from ChR2 expression in glutamatergic terminals from non-preBotC neurons and perhaps incomplete light activation of deeper neurons (off target effects). In particular there is concern that the post-stimulus inhibition appears to be a significant effect that is not replicated by the model and suggests that assumptions about ChR2 in the model require greater scrutiny. Absent further development of the model to encompass this phenomenon the claim that the experiments fully confirm model predictions should be scaled back.

Possible non-uniform and off-target expression of ChR2 are discussed on page 31, lines 22–23, lines 21–23 on page 32, and lines 1–7 of the Discussion. In response to the concerns of off-target ChR2 expression the following text was added to the Discussion (page 31, lines 22–23 and page 32, lines 1–7) (line numbers are those in the highlighted version of the resubmission):

“We also note that the transgenic strategy used here may result in expression of ChR2 in glutamatergic terminals and fibers of passage from non-preBötC neurons, which could result in off-target effects. However, as discussed above, we have calibrated the ChR2 model to match the levels of depolarization at a given laser power that we have measured experimentally at the neuronal level. These measurements represent the total amount of neuronal depolarization which would include any contributions from photostimulated glutamatergic fibers and terminals. Thus, our ChR2 model and simulated photostimulation experiments adequately account for all sources contributing to neuronal depolarization so that such off-target effects related to ChR2 expression, if present, would not impact the conclusions of this study.”

Also, the post stimulus inhibition of preBötC burst frequency is noted in the Results section relating to Figure 3 on page 9, lines 3–5. Failure of the model to capture the post-stimulus inhibition of the preBötC burst frequency is discussed on page 33,lines 18–23 and page 34, lines 1–3, and as we now note in the text is likely due to activity dependent activation of the Na^+^/K^+^ ATPase pump (see Pichon *et al.*, *Journal of Neuroscience*. 2017 and Pichon *et al.*, *Current Biology* 2018). Inclusion of the Na^+^/K^+^ ATPase pump and/or other biophysical mechanisms proposed to shape preBötC inspiratory bursts are beyond the scope of the current study and are therefore left for future investigations (also acknowledged on pp. 33 and 34 of Discussion).

We note that we do not claim that the “experiments fully confirm model predictions”. In the abstract lines 6 and 10 we say that the three specific predictions of the model were confirmed by the experiments which we view as a reasonable and accurate statement given the results presented.

2. The authors’ attempts to resolve the conflicting data for the InaP necessity hypothesis in the Discussion overlooked experimental details in papers that counter the authors arguments. For example, Pace et al., (2007) showed that bilateral microinjection of riluzole or low concentrations of TTX into preBötC failed to stop the rhythm and that the pharmacological effects of these blockers could be explained by their effects on raphe excitability, which provides tonic excitatory drive to the preBötC. The authors propose that these conflicting results can be explained by differences in slice thickness and incomplete pharmacological penetration; however, the Pace paper specifically addressed this issue by microinjecting the drugs 100 μm below the surface. This could, of course, still lead to differences in penetration and there are still differences in the amount of network in the slice, but the discussion of thick vs. thin omits needed details here, and these effects can be addressed experimentally, in future. Absent such future experiments the conclusion that INaP is essential for must be removed and replaced by a more balanced conclusion.

We thank the reviewer for highlighting this point, as addressing these issues in more detail strengthens the manuscript. The text on page 38–40 has been updated to provide a more complete and balanced discussion regarding conflicting data from previous pharmacological I_NaP_ blocking studies and the specific details and hypothesis/conclusion of Pace *et al.,* (2007) are discussed in detail on page 38, lines 1–20 and page 39, lines 13–23. Specifically, the results presented in Pace *et al.,* (2007) suggest that bath application of RZ or TTX does not compromise the fundamental mechanism(s) of preBötC rhythmogenesis but rather the level of neuronal excitability via off-target effects in the raphe obscurus. However, these results have not been reproduced and are even refuted in a follow-up study, Koizumi and Smith (2008). More importantly, the current study is a direct test of the hypothesis presented in Pace *et al.,* (2007). If, as suggested by Pace *et al.,* (2007), bath application of TTX or RZ impacts the inspiratory rhythm by reducing preBötC excitability rather than by affecting the essential mechanism(s) of rhythm generation, then increasing preBötC excitability via optogenetic stimulation should restart the rhythm even after complete I_NaP_ blockade. Our results show that, to the contrary, the preBötC is incapable of generating rhythmic output after complete I_NaP_ block even under optogenetic stimulation (Figures 4 and 5), demonstrating that I_NaP_ is essential for preBötC rhythm generation in this reduced in vitro preparation.

We acknowledge that this study cannot definitively prove the necessity of I_NaP_ for rhythm generation in the isolated preBötC. However, in the current study, the isolated preBötC in this in vitro preparation is incapable of generating rhythmic activity (see Figures 4 and 5) after application of TTX or riluzole at concentrations that completely block I_NaP_ at least for the neurons recorded from (TTX: 20 nM, RZ: 20 µM, See Figure 4—figure supplement 1). Therefore, the statement (page 38, lines 17–20) that I_NaP_ is essential for rhythm generation in the reduced in vitro preparation used in this study is consistent with the computational and experimental results presented in the present paper. Additionally, we note (page 40, lines 11–17) that the role of I_NaP_ in more intact preparations (thick in vitro slices, in situ and in vivo *preparations*) appears to be limited as I_NaP_ block in these preparations do not disrupt rhythm generation (Smith *et al.*, 2007; Rubin *et al.*, 2009b; Rubin and Smith, 2019). Interestingly, predictions from previous computational modeling suggest that I_NaP_ may also be essential under some network conditions in the more intact respiratory network (Phillips and Rubin, 2019; Rubin and Smith, 2019), and these hypotheses remain to be tested in future experiments (Discussion on page 40 referenced above).

3. Reviewer #3 asked for more detail about individual bursting neurons and their firing profiles because the relative expression of gNaP is important for endogenous bursting neurons. So, the authors would have a stronger argument if they included gNaP/Cm (as was done in the Koizumi et al., (2008)) and then showed what happens to those individual neurons when INaP and I_CAN are blocked. That would speak to questions re: variability in rhythm and give an idea of just how much InaP is present in the neurons recorded.

We thank the Editor and the Reviewer for raising this point. We now include the available gNaP/C_m_ data in the Results section (page 10, lines 17–22)**.** We analyzed I_NaP_ attenuation by the pharmacological blockers in 14 preBötC inspiratory glutamatergic neurons as shown in Figure 4—figure supplement 1. In this dataset, there were five endogenous burster inspiratory neurons, which have significantly higher g_NaP_/Cm (105.9 ± 6.5 pS/pF) compared to non-endogenous burster inspiratory neurons (49.6 ± 3.5 pS/pF) (p<0.05 by non-parametric Kolmogorov-Smirnov test), comparable to the results from the previous study of neuronal properties performed in neonatal rat slices in vitro (Koizumi and Smith, 2008). However, I_NaP_ attenuation by the blockers was not significantly different in these two groups (TTX; p=0.464 at 5 nM, p=0.982 at 10 nM, p=0.857 at 20 nM, RZ; p=0.933 at 5 µM, p=0.4 at 10 µM, p=0.933 at 20 µM by non-parametric Kolmogorov-Smirnov test). As we found in neonatal rat preBötC neurons in the earlier study (Koizumi *et al.*, 2008), all of the neonatal mouse inspiratory neurons that we examined had I_NaP_, and the data indicated above should provide the information on differences between endogenous and non-endogenous burster inspiratory neurons. We have not performed cellular-level experiments on these two different types of neurons during block of I_CAN_, which would be important to do in future experiments, but we believe that this is beyond the scope of the present work. Our focus in the present studies was primarily on the important problem of analyzing population-level perturbations and model predictions for the excitatory preBötC neurons, which we could accomplish uniquely with the optogenetic strategy employed.

4. Please address the statistical issues brought up by the reviewers.

As suggested by the reviewers, we have re-analyzed statistical significance throughout as now presented in Results with non-parametric Wilcoxon matched-pairs signed rank test or Kolmogorov-Smirnov test when comparing two groups, and two-way ANOVA tests for comparing multiple groups in conjunction with post hoc Tukey’s HSD tests for pairwise comparison. We have updated the Results section, and also methods section accordingly. Please note that the new results are not different from the previous results in the original submission in terms of statistical significance.

5. Please provide the missing details of the model requested by the reviewers.

Missing model details have been added and detailed below in response to individual reviewers.

6. Each reviewer provides detailed comments that will supplement and expand this summary.

Responses to each comment are detailed below.

Reviewer #1 (Recommendations for the authors):In the current study, Phillips at al., experimentally tested three prediction that emerged from a previously published computational model (eLife 2019, 8:e41555): (1) the blockade of ICAN and InaP produces opposite effects on preBötC rhythmic activity; (2) ICAN is essential for preBötC rhythmogenesis; and (3) ICAN is key for generating the amplitude of respiratory rhythmic output. To do so, the authors used optogenetic/pharmacologic stimulation of the preBötC on mouse brainstem slices. These three predictions are, to a large extend, demonstrated with the new provided experimental data. Globally, the new findings reported by Phillips and colleagues foster our understanding on the elusive mechanisms that underline the generation of the respiratory (more precisely inspiratory) rhythm in mammals, which are of great interest for researchers working in respiratory physiology.Comments1. I have no problems with the model simulations/predictions nor with the findings of this carefully done work. Nevertheless, in my view, it is written in a highly technical manner that is not accessible to neuroscientists working on areas distinct to electrophysiology/computational odelling, which might preclude the full understanding of this very interesting study. Therefore, I would suggest to the authors to work a little bit on making this study more accessible to the large readership of eLife.

We appreciate this reviewer’s suggestion, although we note that Reviewer #3 characterizes the manuscript as “well-written and clear”. Also, we have written many previous papers for a variety of journals, including *eLife*, and the level of technicality of the writing in this article is no different from those well-received articles. We have revised and updated the Introduction for general interest as best we can for a Research Advance. This is a combined experimental and computational study involving computational and experimental advances, so it is not possible to avoid including the necessary technical aspects of our study. Aspects of general interest including the essential scientific background were elaborated in the original Philips *et al.,* 2019 publication in *eLife*.

2. One aspect that could perhaps be discussed in this work is if Phillips's model could also consider the interconnections existing between left/right preBötC in rhythmogenesis and population activity amplitude. In other words, can the current computational model predict how ipsilateral changes in ICAN and INaP might alter population activity amplitude and rhythmogenesis on the contralateral preBötC?

Analysis of the interactions between left/right preBötC networks is beyond the scope of the current study. However, the degree of synchrony between network bursts in the two populations will be highly dependent on the strength and organization of synaptic interactions between these two regions. Investigation of left/right preBötC interactions are therefore left for future computational and experimental investigations.

Reviewer #2 (Recommendations for the authors):Connectivity has been shown to be an important parameter in preBötC dynamics and was explored in the previous publication of this model, but the connectivity matrix/synaptic parameters are not described in this text and should be included.

The synaptic connectivity probability and weights are given in Table 1 in the Methods section of the results. A connection probability of 1 (i.e., all-to-all connectivity) was used in the simulations presented in Figures 1 and 8. As shown in the previous study (Phillips *et al.*, 2019) the model behavior is not qualitatively impacted by changes in the synaptic connectivity probability provided that the synaptic strength remains constant (number of neurons*connection probability*synaptic weight = constant). Similarly, the predicted directional shifts in the preBötC network burst frequency/amplitude curves are not impacted by a reduction in the connection probability to 13% as estimated by Rekling *et al.*, *J. Neurosci*. (2000); see the new Figure 1—figure supplement 1.

Data from the model showing how optogenetic stimulation in the model compares to experimental results, particularly with respect to poststimulation membrane polarization and network effects, e.g., inhibition of rhythmicity following stimulation, should be presented.

We now specifically mention that the model cannot capture this feature in the revised Discussion (starting on p.33, line 18). Also, note that this inhibition is not equally strong in all experiments as discussed; for example, it is considerably less strong in Figure 7B.

In "Model Tuning", changed parameters are stated to be marked in red, but there is no red text; although, some values/terms do appear to be slightly bolded. Please use a clearer mark that is color-blind friendly to indicate updated parameters.

The parameters that are new, different, or restricted compared to the original model tuning in Phillips *et al.,* (2019) are marked in a red font.

The rationale for using power spectrum analysis over analyzing the amplitude and frequency of preBötC activity is unclear. The physiological relevance of power in higher harmonic frequencies should be explained.

We have added the zero frequency and amplitude data at the various levels of preBötC photostimulation after complete block of I_NaP_ with 20 nM TTX and 20 µM RZ in Figure 4. The rationale for employing power spectrum analyses is to test if any rhythmic structure remained in the recorded signals after complete I_NaP_ blockade. As we discussed (p. 35, para. 2), we were testing for rhythmogenic mechanisms that do not depend on I_NaP_ after fully blocking this conductance. Specifically, as discussed, these experiments were designed to determine if there are additional rhythmogenic mechanisms inherent within the excitatory network that emerge at various levels of tonic activity/excitation, controlled in our experiments by sustained bilateral photostimulation, after blocking I_NaP_. This rationale is also now made clearer in the Results section (p. 13, lines 5–8). We also note that the higher harmonic frequencies given by the Fast Fourier Transform of the rectified preBötC activity signal reflect the periodicity and (non-sinusoidal) shape of the bursts of population activity.

When discussing TRPM4 and ICAN, Picardo et al., 2019 PLOS Biology should be cited and discussed.

We have added and discussed this reference (p. 36, line 17 where we discuss the role of TRPM4 and ICAN) as recommended.

[Editors' note: further revisions were suggested prior to acceptance, as described below.]

Reviewer #2 (Recommendations for the authors):In this revised manuscript, Phillips et al. respond to many of the reviewers’ concerns; however, there remain a few issues that should be addressed.From Essential Revisions1. Regarding limitations of the ChR2 experiments, the authors should mention and discuss the possibility that optogenetic stimulation of glutamatergic terminals may synaptically activate preBötC inhibitory neurons, thus altering excitation-inhibition balance (Ashhad and Feldman 2020). Could this mechanism also explain the post-stimulus inhibition?

In this regard, in the section “Extensions and limitations of the model”, we have added a statement on p. 31 about glutamatergic terminals, “including on preBötC inhibitory neurons that are proposed to interact with the excitatory neurons (e.g., Ausborn *et al.*, 2018; Ashhad and Feldman, 2020)”

2. At the end of the Discussion, in the Summary, and in the author's response, the authors state that their results support the statement: "INaP is essential for rhythm generation in the reduced in vitro preparation used in this study." A similar qualifier, like adding "in vitro", should be placed wherever a statement about INaP being essential is made, especially in the abstract, second to last paragraph of Introduction, and first two paragraphs of the Discussion.

As suggested, we have further emphasized that the results apply to circuit operation in vitro in the Abstract and elsewhere, although this qualification was in the original manuscript and made clear in the Introduction and Discussion.

Reviewer #3 (Recommendations for the authors):The authors have addressed the majority of my comments. My only remaining concern is their unwillingness to assess variability in their data since this seems a trivial "ask"---particularly if they choose perhaps the simplest metric of variability, coefficient of variation. Nonetheless, I still feel the manuscript is of value and advances the field and the authors have addressed the majority of concerns for each of the reviewers.

We are not unwilling to assess variability of the data. There are multiple datasets of different types presented in the paper, and in the key summary data graphical representations, we provide the ± SEM, which is an appropriate measure of reliability and reflects the standard deviation for the means. The reviewer suggests that we provide the CV, but it is unclear to us in which datasets of the different types this additional metric of variability should be indicated, and what this would actually add to the representation of the data. The data analyses presented were designed for comparisons to model predictions. As explained in our original response to this reviewer, we think that our normalization of some of the data, while tending to make the data appear tighter, was the appropriate way to allow for directional comparisons to the model predictions under the various experimental conditions.